# Dynamics of accessible chromatin regions and subgenome dominance in octoploid strawberry

Chao Fang[1], Ning Jiang[2,3,4], Scott J. Teresi [2,4], Adrian E. Platts [2], Gaurav Agarwal[1], Chad Niederhuth[1,3], Patrick P. Edger [2,3,4] ✉ & Jiming Jiang [1,2,3] ✉

Subgenome dominance has been reported in diverse allopolyploid species, where genes from one subgenome are preferentially retained and are more highly expressed than those from other subgenome(s). However, the molecular mechanisms responsible for subgenome dominance remain poorly understood. Here, we develop genome-wide map of accessible chromatin regions (ACRs) in cultivated strawberry ($2n = 8x = 56$, with A, B, C, D subgenomes). Each ACR is identified as an MNase hypersensitive site (MHS). We discover that the dominant subgenome A contains a greater number of total MHSs and MHS per gene than the submissive B/C/D subgenomes. Subgenome A suffers fewer losses of MHS-related DNA sequences and fewer MHS fragmentations caused by insertions of transposable elements. We also discover that genes and MHSs related to stress response have been preferentially retained in subgenome A. We conclude that preservation of genes and their cognate ACRs, especially those related to stress responses, play a major role in the establishment of subgenome dominance in octoploid strawberry.

Polyploidy is a prevalent and recurrent evolutionary force that has contributed for novelty and driven net diversification rates across eukaryotes, and is especially common among plants[1,2]. For example, all angiosperms have undergone at least one polyploidization event during their evolutionary history[3]. Polyploids can be classified into two types based on the origin of their multiple chromosome sets: autopolyploids, which result from whole genome duplication (WGD) of a single diploid progenitor species, and allopolyploids, which arise from hybridization of two or more distinct diploid progenitor species combined with WGD[4]. Rapid and extensive genetic and epigenetic changes often occurred during the establishment and stabilization of newly established allopolyploids[5]. In some allopolyploids, genes from one subgenome were preferentially retained or achieved a higher level of expression than those from other subgenomes[6,7], which is referred to as "subgenome dominance" and has been documented in an increasing number of allopolyploids[6–17]. A classic example of subgenome dominance has been established in maize. A WGD occurred in maize roughly 5–12 million years ago (MYA), with nearly two-thirds of the duplicated gene pairs have since returned to one copy[18,19]. Interestingly, the two ancestral subgenomes have undergone uneven loss of the duplicated genes[20]. The dominant subgenome in maize has lost significantly fewer genes and exhibits higher levels of gene expression[7,16] – which are the two primary characteristics used to define subgenome dominance in allopolyploids[21]. However, the molecular mechanisms of subgenome dominance remain poorly understood.

Although subgenome dominance has been documented in a number of plant species, not all polyploids exhibit subgenome dominance[22–25]. The lack of subgenome dominance was thought to be skewed toward autopolyploids or polyploids with highly similar

[1]Department of Plant Biology, Michigan State University, East Lansing, MI 48824, USA. [2]Department of Horticulture, Michigan State University, East Lansing, MI 48824, USA. [3]Michigan State University AgBioResearch, East Lansing, MI 48824, USA. [4]Genetics and Genome Sciences Program, Michigan State University, East Lansing, MI 48824, USA. ✉e-mail: edgerpat@msu.edu; jiangjm@msu.edu

subgenomes[26–28]. Thus, the pre-existence of genetic differences among the progenitor species may be driving subgenome dominance in certain allopolyploids. Transposable elements (TEs) content is one genomic feature known to often be highly variable within and among closely related plant species[29–31]. Furthermore, it is well known that TEs can impact the expression of neighboring genes genetically and/or epigenetically[32–35]. The genomic shock from a polyploidization event may induce proliferation of TEs residing in the progenitor genomes, which may further differentiate the TE abundance among different subgenomes. Therefore, it was not surprising that low TE abundance has been previously associated with a dominant subgenome in several allopolyploid species[7,11,36–39]. In other words, several previous studies have shown that the higher expression level of homoeologs encoded on the dominant subgenome is generally inversely correlated with the density of methylated transposable elements compared to the homoeologs on the submissive subgenomes[9,11,27,28,37,40–42]. To complement these findings, previous studies of polyploids which lack any evidence for subgenome dominance have reported similar TE densities near homoeologs between subgenomes[22,23]. While differences in TEs have been studied in various allopolyploids, other genomic features, including variation in noncoding regulatory regions, as a potential driver of observed homoeolog expression differences, remain to be investigated in allopolyploids[28].

Cultivated strawberry (*Fragaria × ananassa*) is an interspecific hybrid of two wild octoploid strawberry species, *Fragaria virginiana* and *Fragaria chiloensis*[43]. These octoploid strawberries share a common ancestor that was formed from the hybridization and polyploidization involving four diploid progenitor species more than one MYA[44]. Octoploid strawberry still contains a complete set of homoeologous chromosomes from all four diploid progenitors. Among the four subgenomes (A, B, C, D), the A subgenome, which was contributed by an ancestor most closely related to *Fragaria vesca* ssp. *bracteata*, encodes a significantly greater amount of more highly expressed homoeologs compared to the submissive B/C/D subgenomes[39]. In addition, the dominant A subgenome has the lowest TE densities near genes compared to the other three subgenomes[39]. The dominant A subgenome also lost significantly fewer genes compared to the submissive subgenomes[39], similar to patterns reported for Arabidopsis[6] and maize[7]. Therefore, octoploid strawberry serves as a useful model system to further study the underlying mechanisms that contribute to subgenome dominance.

While there is community consensus about which chromosomes are assigned to the A subgenome, there are some inconsistencies on which chromosomes should be assigned to the B/C/D subgenomes[45–49]. There is strong and consistent support for six of seven chromosomes for the B subgenome, the only exception being chromosome 6 from a single study[50]. For chromosome assignments to the C and D subgenomes, there is consistent support for only chromosomes 1, 4 and 7[39,48,50,51]. The incongruence of chromosome assignments from these studies stems in part due to differences in their methodological approaches. Nevertheless, potential issues associated with chromosome (mis)assignments between submissive subgenomes can be addressed by combining and averaging across the submissive subgenomes[37,39]. Lastly, regarding the hybridization order, there are consistent results that the A subgenome, which was the maternal donor, hybridized with the hexaploid ancestor to form the octoploid[39,48]. Based on the analyses of repeat content, Session and Rokhsar (2023) proposed a model by which the tetraploid was formed by the hybridization of the diploid progenitors of C and D, which subsequently hybridized with the B subgenome donor to form the ancestral hexaploid.

In this study, we develop a genome-wide map of accessible chromatin regions (ACRs) in octoploid strawberry using MNase-hypersensitivity sequencing (MH-seq)[52,53]. Each ACR is identified as an MNase hypersensitive site (MHS). We use the MHS dataset to explore differences between the previously identified dominant A and submissive B, C and D subgenomes. MHS and gene expression datasets are used to further investigate subgenome dominance based on turnovers of the DNA sequences associated with MHSs combined with differences in expression abundance between homoeologous genes. We demonstrate that the dynamics of the ACRs, including sequence variation and mutation, loss of ACRs, and transposon interruptions, play an important role in observed subgenome dominance in strawberry.

## Results

### Accessible chromatin regions in strawberry genome
To identify ACRs in the strawberry genome, we developed MH-seq libraries[54] using chromatin isolated from leaf tissue of octoploid strawberry cultivar 'Royal Royce' (RR). The MH-seq reads from two biological replicates were mapped to a high-quality RR reference genome[55]. We obtained 63 and 74 million uniquely mapped reads from the two biological replicates, respectively. A strong correlation (r = 0.88) was found between the two replicates (Supplementary Fig. 1a). Therefore, the MH-seq reads from the two libraries were combined for identification of MHSs in the genome. We identified 96,841 MHSs using F-seq[56]. The MHSs were classified into four categories: (1) "Upstream MHSs", which are located within 1 kb upstream of a transcriptional start site (TSS); (2) "gene body MHSs", which are located within 5'UTR, exon, intron, or 3'UTR; (3) "downstream MHSs", which are located within 1 kb downstream of a transcriptional terminal site (TTS); (4) the remaining DHSs are designated as "intergenic MHSs". Nearly 68% MHSs were located within ±1 kb regions of annotated genes, including those in the gene bodies (Supplementary Fig. 1b).

### MHSs associated with the four subgenomes
We previously demonstrated that subgenome A shows gene expression dominance over the B, C, and D subgenomes in octoploid strawberry in all examined organs including leaves[39]. In this study, we are following the chromosome and subgenome assignment for octoploid strawberry as previously proposed[46]. All genes assigned to a particular chromosome and/or subgenome are treated as coming from a single diploid progenitor species. We generated RNA-seq data from RR leaf tissue and classified genes into expressed (TPM > 0) and non-expressed genes (TPM = 0). Subgenome A contained a significantly lower proportion (A: 33.5% vs. BCD: ~35.0%, $p < 1.8e-2$, chi-square test) of non-expressed genes and a significantly higher proportion (A: 66.5% vs. BCD: ~65.0%, $p < 1.8e-2$, chi-square test) of expressed genes compared to the B, C, and D subgenomes (Fig. 1a).

Next, we assigned each MHS to one of the four subgenomes. Subgenome A contained the greatest number of MHSs (Fig. 1b, Supplementary Fig. 1c, Supplementary Table 1). To account for gene content differences among the subgenomes, we normalized the number of MHSs by the number of genes within each subgenome. Subgenome A had a significantly higher ($p < 0.01$, one-way ANOVA with Games-Howell *post-hoc* test) number of MHSs per gene than the other three subgenomes (Fig. 1c). Furthermore, the average length of MHSs per gene associated with subgenome A was significantly higher ($p < 0.01$, one-way ANOVA with Games-Howell *post-hoc* test) than those associated with the B/C/D subgenomes (Fig. 1d). These results indicate that the difference in MHS content associated with subgenome A may be an important factor to the dominance of subgenome A in octoploid strawberry.

To further validate the dominant MHS features associated with subgenome A, we compared the MHS numbers from A subgenome with those from combined C/D or combined B/C/D subgenomes to minimize the impact from potential misassignments of individual chromosomes to C and D subgenomes (Supplementary Fig. 2). We grouped subgenome C and D together and calculated the average number of MHS per gene by dividing the total number of MHSs from

subgenome C and D to the total number of genes in these two subgenomes, we observed that the numbers of MHSs per gene are comparable between subgenome B and the combined subgenomes C/D, and both are significantly lower than that of subgenome A (Supplementary Fig. 2a). Similar results were also obtained when subgenomes B, C, and D were combined (Supplementary Fig. 2b). Due to the chromosome assignments for C and D differ for chromosomes 2, 5 and 6 in recent studies[48,49,57], we swapped these chromosomes between the C and D subgenomes and compared the average MHS number across subgenome A, B, and the newly created subgenomes C and D. We found that the average MHS number of subgenome A is still significantly higher than that of the B subgenome and the newly created C or D subgenomes (Supplementary Fig. 2c).

Finally, we investigated the differences in the average number and length of MHS per gene for each chromosome (Supplementary Fig. 3). This revealed that chromosomes 1 through 6 had patterns consistent with subgenome-wide patterns, with those assigned to the dominant A subgenome having a higher average number of MHSs per gene and longer average MHS length per gene compared to the three submissive subgenomes. The differences were significant ($p < 0.01$, one-way ANOVA with Games-Howell *post-hoc* test) for each pairwise comparison between subgenome A chromosome against the homoeologous chromosomes assigned to the other subgenomes, except for between chromosome 1A and 1C (average MHS length per gene only), chromosome 3A and 3D (both number of MHS per gene and average MHS length per gene), and chromosome 6A and 6B (both number of MHS per gene and average MHS length per gene). The only notable exception is for chromosome 7 comparisons: chromosome 7D has the highest average number of MHSs per gene and longer average MHS length per gene (Supplementary Fig. 3). However, this difference between 7A and 7D is not significant for either MHS length or number,

but 7D is significantly different from 7B and 7C and 7A is significantly different from 7C ($p < 0.01$, one-way ANOVA with Games-Howell *post-hoc* test). The observed differences for these chromosomes may in part be explained by homoeologous exchanges that have occurred between subgenomes[39,46,49,51].

For example, one estimate of homoeologous exchanges in octoploid strawberry, based on phylogenetic analyses[39], suggested that chromosome 7D had the largest amount of subgenome A content compared to all other submissive subgenome chromosomes. Thus, this may be the reason for the lack of observed statistical difference between 7A and 7D. Furthermore, variation in parental subgenome dosage is well documented in many allopolyploid species due to homoeologous exchanges and whole chromosome replacement, as observed in *Brassica napus*[58]. However, the aforementioned phylogenetic analyses in strawberry were conducted using assembled transcriptomes from diverse diploids, and thus provides a sparse estimate of genome wide patterns and lacks precise prediction of boundaries for individual homoeologous exchanges[39]. Future phylogenetic studies are needed that utilize entire assembled genomes of extant relatives of each subgenome to improve estimates of the homoeologous exchange landscape in octoploid strawberry, and to further evaluate the patterns of MHS sequences uncovered in this study.

Given the comparable results observed for both the average number of MHS per gene and the average MHS length per gene, irrespective of employing the subgenome assignment based on the RR reference genome[55], or combining subgenomes B/C/D, or swapping chromosomes 2, 5 and 6 between subgenomes C and D, or even segregating each chromosome for comparison, we performed all subsequent analyses using the subgenome assignment based on the RR reference genome[55].

## Correlation between chromatin accessibility and transcriptional divergence of homoeologous genes

We analyzed the association of MHS variation with transcriptional divergence of homoeologous genes. We first identified syntenic homoeologous genes in each subgenome. More than 72% (19,796/27,398) of genes identified in subgenome A have a homoeolog in at least one of the three other subgenomes with 66% (13,115/19,796) of them having a homoeolog in all three subgenomes (Fig. 2a). We investigated the number of genes from subgenome A that show higher expression levels than their homoeologous genes encoded in the submissive subgenomes. For each homoeologous gene pair, one copy was defined to have an elevated expression if it was expressed at least 1.5-fold higher than the other copy. More than 34% (6,824/19,796) of subgenome A genes were more highly expressed than at least one of their B/C/D homoeologs. The proportion of more highly expressed genes in the B, C, and D subgenomes was significantly lower (-29%; $p < 2.2e-16$, chi-square test) than that of subgenome A (Fig. 2b). Similarly, we investigated the numbers of genes from subgenome A showing 1.5-fold lower expression than their homoeologs. Only 22% (4387/19,796) of genes from subgenome A showed lower expression than at least one of their B/C/D homoeologs. The proportion of similarly lower expressed genes in the B, C, and D subgenomes was significantly higher (-30%; $p < 2.2e-16$, chi-square test) than that of subgenome A (Fig. 2c).

Next, we investigated the chromatin accessibility differences between the dominant subgenome A genes associated with higher and lower expression levels compared to homoeologous genes encoded in the submissive subgenomes. The 6824 dominantly expressed genes from subgenome A were divided into three groups: (1) genes expressed higher than the homoeologs from all three submissive subgenomes (778); (2) genes expressed higher than the homoeologs from two of the three submissive subgenomes (2058); (3) genes expressed higher than the homoeologs from only one of the three submissive subgenomes (3988). Among all three groups, the 5' region of the

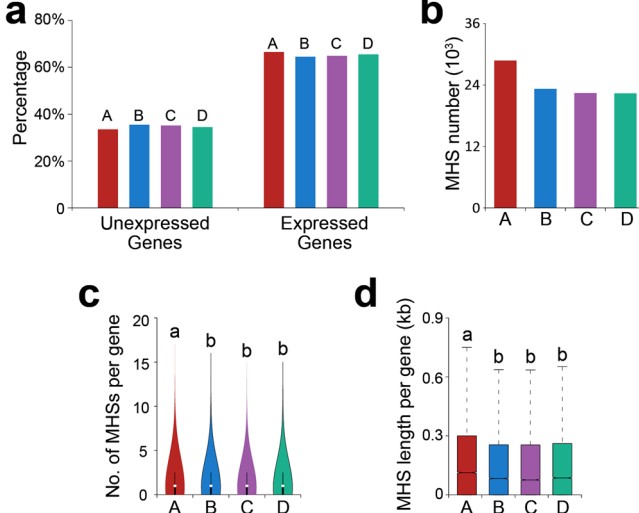

**Fig. 1 | Gene expression and chromatin accessibility associated with the four subgenomes (A, B, C, D) in strawberry. a** Percentage of expressed and non-expressed genes in the four subgenomes. Subgenome A has a lower percentage of non-expressed genes and a higher percentage of expressed genes (A vs. B: $p = 2.4e-6$; A vs. C: $p = 1.2e-4$; A vs. D: $p = 1.8e-2$, two-sided chi-square test). **b** A total number of MHSs in each subgenome. **c** Average number of MHSs per gene in each subgenome ($n_A = 27,398$, $n_B = 25,461$, $n_C = 24,505$, $n_D = 23,911$). **d** Average MHS length per gene in each subgenome ($n_A = 27,398$, $n_B = 25,461$, $n_C = 24,505$, $n_D = 23,911$). The lower and upper boundaries of each box indicate 25th and 75th percentile, the center line indicates the median, and the whiskers extend to 1.5× IQR. In (**c**, **d**), the total number of MHSs and MHS length were normalized in the four subgenomes by dividing to the total number of genes from each subgenome. Means that do not share a letter are significantly different ($p < 0.01$, one-way ANOVA with Games-Howell *post-hoc* test). Source data are provided as a Source Data file.

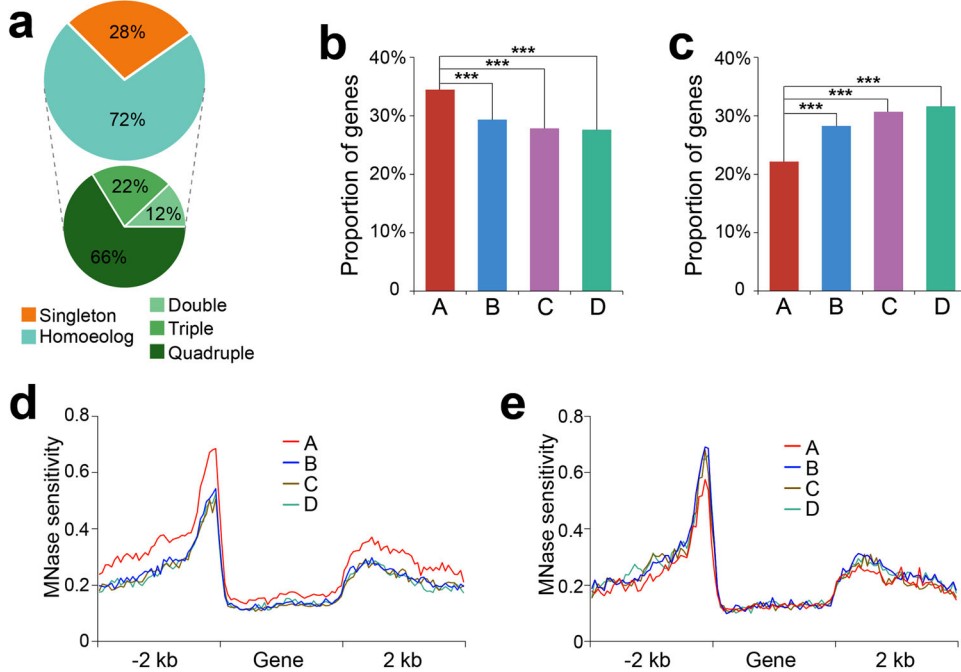

**Fig. 2 | Chromatin accessibility and its impact on transcriptional divergence of homoeologous genes. a** The proportion of genes from subgenome A with or without homoeologs identified in B/C/D subgenomes. Singleton indicates genes that do not have a homoeolog from any of the other three subgenomes. Homoeolog indicates genes that have homoeologs from at least one of the other three subgenomes. Double, triple, and quadruple indicate genes that have one, two, and three homoeologs, respectively. **b** Proportion of gene from subgenome A showing higher expression than at least one of its homoeologs (A vs. B, C or D: $p < 2.2\mathrm{e}{-16}$). **c** Proportion of gene from subgenome A showing lower expression than at least one of its homoeologs (A vs. B, C or D: $p < 2.2\mathrm{e}{-16}$). **d** MH-seq profiles of the highly expressed genes from subgenome A (n = 778) and their homoeologs from B, C, and D subgenomes. **e** MH-seq profiles of the lowly expressed genes from subgenome A (n = 397) and their homoeologs from B, C, and D subgenomes. \*\*\*$p < 0.001$, two-sided chi-square test. Source data are provided as a Source Data file.

subgenome A genes showed a higher chromatin accessibility level than their homoeologs from the B/C/D subgenomes (Fig. 2d, Supplementary Fig. 4). Similarly, the 4387 lowly expressed genes from subgenome A were also divided into three groups: (1) genes expressed lower than the homoeologs from all other three subgenomes (397); (2) genes expressed lower than the homoeologs from two of the three submissive subgenomes (1064); (3) genes expressed lower than the homoeologs from only one of the three submissive subgenomes (2926). Among all three groups, the 5′ region of the subgenome A genes showed a lower chromatin accessibility level than their homoeologs (Fig. 2e, Supplementary Fig. 5). These results showed that the MHS variation is potentially a major contributor to observed transcriptional divergence of the homoeologous genes in octoploid strawberry.

**Divergence of the DNA sequences associated with MHSs**

We identified homoeologous sequences from at least one of the three (B/C/D) subgenomes for more than 87% (25,149/28,798) of subgenome A MHSs (Fig. 3a, Supplementary Table 1), which were termed "syntenic MHSs". For every pair of syntenic MHSs, we calculated the number of MH-seq reads in the MHS and its homoeolog, which were used to measure the differential MNase sensitivity of the pair of syntenic MHSs. We found that 23% (5724/25,149) of syntenic MHSs in subgenome A showed higher MNase sensitivity (hi-MHSs) and 10% (2391/25,149) of syntenic MHSs in subgenome A showed lower MNase sensitivity (lo-MHSs). We further identified hi-MHSs and lo-MHSs for each of the other three subgenomes. Subgenome A showed a significantly higher percentage of hi-MHSs ($p < 2.2\mathrm{e}{-16}$, chi-square test) and a significantly lower percentage of lo-MHSs ($p < 2.2\mathrm{e}{-16}$, chi-square test) compared to the B, C, and D subgenomes (Fig. 3b, c), indicating increased MNase sensitivity and chromatin accessibility across subgenome A at syntenic MHSs.

We hypothesized that DNA sequence variation and/or mutation may have resulted in loss or alteration of the MNase sensitivity of the MHS-associated genomic regions between the four subgenomes. To test this hypothesis, we identified 2034 hi-MHSs from subgenome A as compared to subgenome B. We randomly selected 2034 pairs of control-MHSs (syntenic MHSs without differential MNase sensitivity) from subgenome A and their subgenome B homoeolog as a control. SNPs (single nucleotide polymorphism) and INDELs (insertion and deletion) were identified between each pair of hi-MHS/control-MHS. The numbers of SNPs/INDELs between hi-MHSs and their homoeologs were significantly greater than those between control-MHSs and their homoeologs (Fig. 3d). Similar results were obtained for 2075/2053 pairs of hi-MHSs from subgenome A and their homoeologs from subgenomes C/D (Supplementary Fig. 6). These results demonstrate that differential MNase sensitivity is associated with greater sequence variation between syntenic MHS sites.

Sequence variation between homoeologous MHSs may originate in two different ways: (1) the variants were inherited from their diploid progenitors or (2) the variants arose after polyploidization (Fig. 3e). We can distinguish these two scenarios for syntenic MHSs by comparing to the sequence of the closest extant relative of each diploid progenitor. For example, there is a "T/C" SNP between a hi-MHS (T) from subgenome A and its homoeolog (C) from subgenome B (Fig. 3e). We identified the syntenic regions of a hi-MHS in *F. vesca* and *F. iinumae*, which are the closest extant relatives of the diploid progenitors of subgenome A and subgenome B, respectively[39,51]. If the subgenome MHS genotype is the same as the diploid genotype, e.g. T in *F. vesca* and C in *F. iinumae*, then this suggests that this SNP arose prior to polyploidization and was inherited from the diploid progenitors ("inherited type"). However, if the subgenome genotype differs from the diploid genotype, e.g. is not T in *F. vesca* or not C in *F. iinumae*, this suggests that the variant may have arisen via mutation following

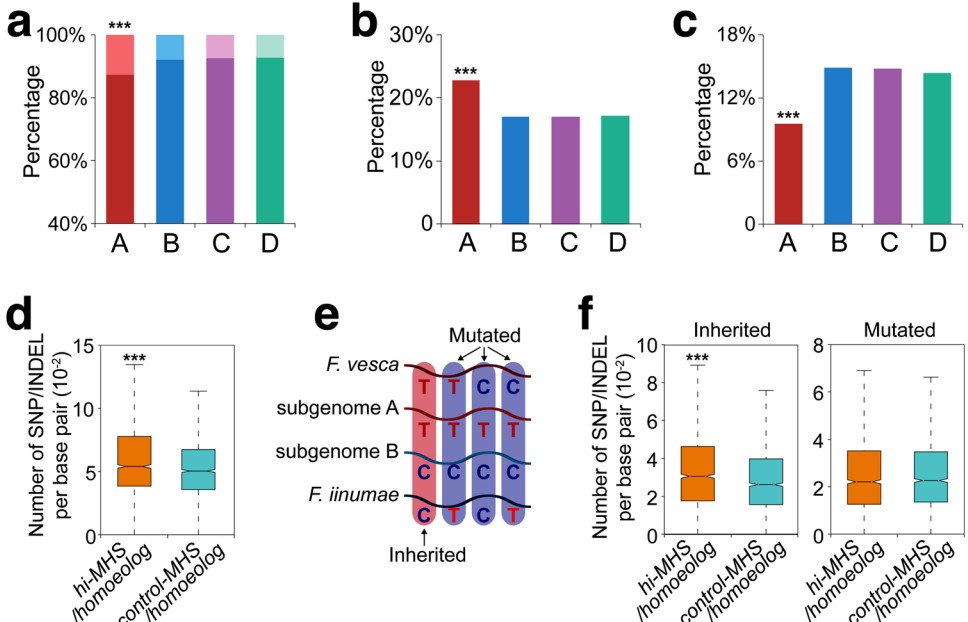

**Fig. 3 | Sequence variation and MHS divergence. a** Proportions of syntenic MHSs and singleton MHSs from the four subgenomes. MHSs with a homoeolog from at least one of the other three subgenomes are referred to as syntenic MHSs, which are indicated by a darker color. MHSs without a homoeolog from any of the other three subgenomes are referred to as singleton MHSs, which are indicated by a lighter color. Subgenome A has a higher percentage of singleton MHSs compared to subgenome B/C/D (A vs. B, C or D: $p < 2.2e{-}16$). **b** Proportion of MHSs showing significantly higher MNase sensitivity than at least one of its homoeologs. Subgenome A has a higher percentage of hi-MHSs compared to subgenome B/C/D (A vs. B, C or D: $p < 2.2e{-}16$). **c** Proportion of MHSs showing significantly lower MNase sensitivity than at least one of its homoeologs. Subgenome A has a lower percentage of lo-MHSs compared to subgenome B/C/D (A vs. B, C or D: $p < 2.2e{-}16$). **d** Numbers of SNPs/INDELs identified in hi-MHS (A, n = 2034)/homoeolog (B, n = 2034) pairs and control-MHS (A, n = 2034)/homoeolog (B, n = 2034) pairs ($p = 3.5e{-}10$). The number of SNP/INDEL was normalized by dividing to the

alignment length between MHS and its homoeolog. **e** Identification of two types (inherited and mutated) of SNP/INDEL between an MHS from A subgenome and its homoeolog from B subgenome. Each row represents four different possibilities of a "T/C" SNP in subgenomes A and B, and in the two diploid progenitor species. Each vertical column illustrates a specific combination of the SNP from four sequence sources, which determine if the SNP was inherited from the diploid progenitor or arose after polyploidization. **f** The number of two types of SNP/INDEL in hi-MHS (A, n = 1726)/homoeolog (B, n = 1726) pairs and control-MHS (A, n = 1726)/homoeolog (B, n = 1726) pairs (Inherited: $p = 3.8e{-}8$; Mutated: $p = 0.64$). The number of SNP/INDEL was normalized by dividing to the overlapped length between an MHS and its homoeolog. The lower and upper boundaries of each box indicate 25th and 75th percentile, the center line indicates the median, and the whiskers extend to 1.5 × IQR in (**d**–**f**). ***$p < 0.001$, two-sided chi-square test (**a**–**c**) and Mann–Whitney $U$ test (**d** and **f**). Source data are provided as a Source Data file.

polyploidization ("mutated type") (Fig. 3e). It needs to be noted that we cannot rule out the possibility that there have been mutations in the extant diploid species following divergence from the subgenome donor. Furthermore, given the rich history of homoeologous exchanges in octoploid strawberry[46], a region encoded on either A or B subgenome may have been contributed by another diploid progenitor species[59].

Using this strategy, we analyzed the 2034 pairs of hi-MHSs from subgenome A and their homoeologs from subgenome B. We were able to identify syntenic regions in *F. vesca* and *F. iinumae* for a total of 1726 pairs of hi-MHSs. We randomly selected 1726 pairs of control-MHSs (syntenic subgenome A/B MHSs without differential MNase sensitivity) and also identified the syntenic regions in the *F. vesca* and *F. iinumae* genomes[60,61]. We found a similar number of variants classified as 'mutated' between the hi-MHS/homoeolog and control-MHS/homoeolog (Fig. 3f). However, the number of variants classified as 'inherited' was significantly higher in hi-MHS/homoeolog pairs than that in control-MHS/homoeolog pairs (Fig. 3f). These results indicate that the inherited variants from the progenitor species, rather than mutations after polyploidization, represent the major contribution to sequence divergence between subgenome A hi-MHSs and their subgenome B homoeologs in octoploid strawberry.

Similarly, we conducted analyses of the hi-MHSs from subgenome A and their homoeologs from subgenomes C/D. *F. iinumae* was used as the diploid genome comparator for both C and D subgenome in the analyses, because of the phylogenetic relationship between *F. iinumae* and the C/D subgenomes in contrast to *F. vesca*[49]. We identified

syntenic regions in *F. vesca* and *F. iinumae* for a total of 1742/1697 pairs of hi-MHSs from subgenome A and homoeologs from subgenomes C/D. We randomly selected the same numbers of pairs of control-MHSs (syntenic subgenome A/C or A/D MHSs without differential MNase sensitivity). We found similar results that the inherited variants from the progenitor species, rather than mutations after polyploidization, represent the major contribution to sequence divergence between subgenome A hi-MHSs and their subgenome C/D homoeologs in strawberry (Supplementary Fig. 7).

## Genes and MHSs specific to subgenome A

Homoeologous gene analysis revealed that 28% (7602/27,398) of subgenome A genes do not have a homoeolog in any of the B/C/D subgenomes (Fig. 2a). These genes were referred to as subgenome A-specific genes. We performed the same analysis and identified 6764, 6526, and 6230 genes specific to B, C, and D subgenomes, respectively. Subgenome A contained a significantly higher percentage of ($p < 0.01$, chi-square test) subgenome-specific genes than the B/C/D subgenomes. Interestingly, GO analysis of the 7602 subgenome A-specific genes revealed enrichment of genes associated with plant responses to various biotic and abiotic stresses (Fig. 4). We performed similar GO analyses of the subgenome B/C/D-specific genes. A total of 74 significantly enriched GO terms related to biological processes (BP) were found from analysis based on the subgenome B-specific genes. Only two of these GO terms, "jasmonic acid and ethylene-dependent systemic resistance" and "response to mechanical stimulus", were related to stress response. Similarly, we identified 8 and 3 significantly

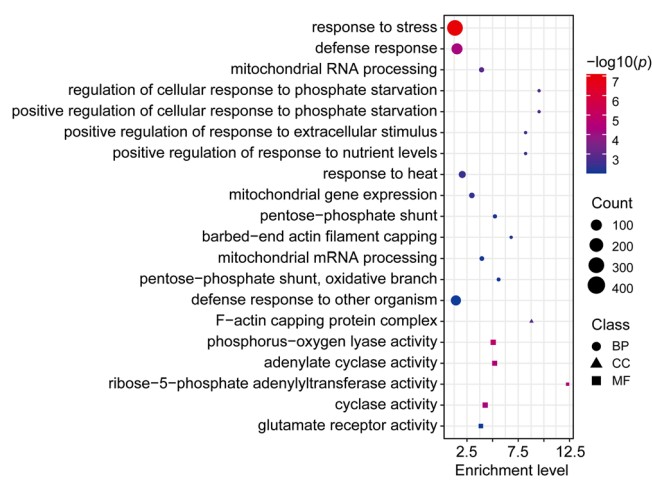

**Fig. 4 | GO terms enriched in subgenome A-specific genes.** BP biological process, CC cellular component, MF molecular function. The significance was assessed using the Fisher exact test (−log10) with adjusted *p*-values calculated by the Benjamini−Hochberg (BH) method. Source data are provided as a Source Data file.

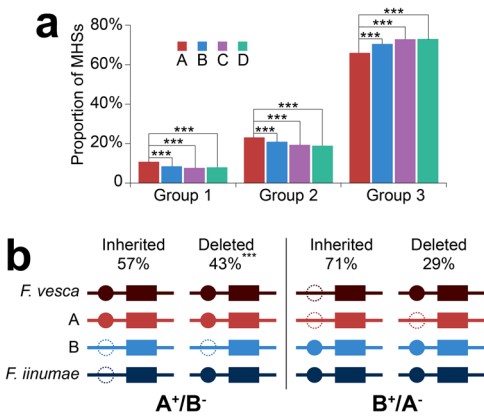

**Fig. 5 | MHSs identified in four subgenomes and the origin of subgenome A/B-specific MHSs. a** The proportions of three groups of syntenic MHSs. Group 1 MHSs lost their homoeologs in two of the other three subgenomes (A vs. B, C or D: $p < 2.2e$ −16); Group2 MHSs lost their homoeologs in one of the other three subgenomes (A vs. B: $p = 1.3e{-}8$; A vs. C or D: $p < 2.2e{-}16$); Group 3 MHSs have homoeologs in all other three subgenomes (A vs. B, C or D: $p < 2.2e{-}16$). **b** Origins of subgenome A/B-specific MHSs. A+/B−: subgenome A-specific MHSs, which do not have a homoeolog in subgenome B. B+/A−: subgenome B-specific MHSs, which do not have a homoeolog in subgenome A. Inherited: a subgenome-specific MHS was inherited from its progenitor species. Deleted: a subgenome A (B)-specific MHS is generated due to loss of its homoeologous B (A) sequence after polyploidization. Solid circles indicate MHSs and rectangles indicate their cognate genes. Dotted circles indicate loss of the homoeologs. ***$p < 0.001$, two-sided chi-square test. Source data are provided as a Source Data file.

enriched BP GO terms from subgenomes C- and D-specific genes, respectively, but none of them were related to stress response.

We also aligned the sequences of all MHSs identified in subgenome A with the genomic sequences of B/C/D subgenomes. Nearly 13% (3649/28,798) of the subgenome A MHSs do not have a homoeolog in any of the other subgenomes and are thereafter referred to as subgenome A-specific MHSs. Nearly 49% (1777/3649) of the subgenome A-specific MHSs were found to be associated with subgenome A-specific genes. We performed the same analysis for the MHSs identified in the B/C/D subgenomes. Subgenome A contained a significantly higher percentage of ($p < 2.2e{-}16$, chi-square test) subgenome-specific MHSs than the three other subgenomes (Fig. 3a).

We searched for enriched DNA sequence motifs within the subgenome A-specific MHSs using the MHSs identified in all four subgenomes as a control. An enriched RVCCMA motif (E-value = 2.1e−13) associated with TEOSINTE BRANCHED 1, CYCLOIDEA, and PCF (TCP) transcription factors (TFs) was identified by MEME software[62]. The TCP TFs have been recognized to play roles in plant response to various abiotic stresses[63–66] as well as in plant development[67,68].

## Loss of MHS-related sequences after polyploidization

We characterized all syntenic MHSs from each subgenome into one of three groups: (1) Group 1: the MHS has a homoeolog in only one of the other subgenomes; (2) Group 2: the MHS has a homoeolog in two of the other subgenomes; (3) Group 3: the MHS has homoeologs in all three other subgenomes. Subgenome A has significantly higher percentages of Group 1 ($p < 2.2e{-}16$, chi-square test) and Group 2 MHSs ($p < 2.2e{-}16$, chi-square test) but a significantly lower percentage of Group 3 MHSs compared to B, C, and D subgenomes (Fig. 5a). In addition, subgenome A contains more subgenome-specific MHSs than subgenomes B/C/D. Taken together, these results suggest that more MHSs identified in subgenome A lost their homoeologous sequences in B/C/D subgenomes.

Similar to our analysis of SNPs/INDELs associated with homoeologous MHSs, lack of syntenic MHS between subgenomes could be due to inherited sequence variation from the progenitor species or could arise after polyploidization (Fig. 5b). We analyzed the subgenome A and B MHSs lacking respective homoeologous subgenome B and A MHSs. We identified 6530 subgenome A MHSs without a subgenome B homoeolog, of which 43% (2813/6530) were found to have homoeologous sequences in *F. iinumae*, suggesting that the homoeologous subgenome B sequences were lost after polyploidization (Fig. 5b). For subgenome B, we identified 3201 MHSs without a

subgenome A homoeolog, but only 29% (943/3201) of these MHSs were found to have homoeologous sequences in *F. vesca*.

We conducted a similar comparative analysis between subgenome A and subgenomes C/D using *F. iinumae* as the diploid comparator. We identified 7328 subgenome A MHSs without a subgenome C homoeolog, of which 52% (3795/7328) were found to have homoeologous sequences in *F. iinumae*. In comparison, 854 of the 2791 subgenome C MHSs (29%) without a subgenome A homoeolog were found to have homoeologous sequences in *F. vesca*. Similarly, we identified 7393 subgenome A MHSs without a subgenome D homoeolog, of which 52% (3874/7393) were found to have homoeologous sequences in *F. iinumae*. In comparison, 891 of the 3091 subgenome D MHSs (30%) without a subgenome A homoeolog were found to have homoeologous sequences in *F. vesca*. Collectively, these results confirmed that fewer MHSs were lost from subgenome A after polyploidization compared to subgenome B/C/D (29% vs 43%, 29% vs 52%, 30 vs 52%, $p < 2.2e{-}16$, chi-square test).

We next investigated the impact on gene expression from the 2813 subgenome A MHSs lacking subgenome B homoeologs. The 2813 MHSs were first filtered using the following three steps: (1) 1718 (61%) MHSs were removed because their cognate genes do not have a homoeologous gene in subgenome B; (2) 249 (9%) MHSs were removed because their cognate genes show no or very low expression in both subgenomes A and B (TPM < 1); (3) 254 (9%) MHSs were removed because their cognate genes are highly diverged compared to their subgenome B homoeologous genes (sequence similarity between homoeologous genes is <80%). This left 622 MHSs associated with 533 pairs of homoeologous A/B genes. We found that 192 genes (36%) from subgenome B showed less than 67% expression levels compared to their homoeologous A genes. Meanwhile, we randomly selected 533 pairs of A/B homoeologous genes as a control, which were not associated with any presence/absence variation of MHSs. Only 24% of the subgenome B genes showed less than 67% expression levels compared to their homoeologous A gene, which is significantly lower

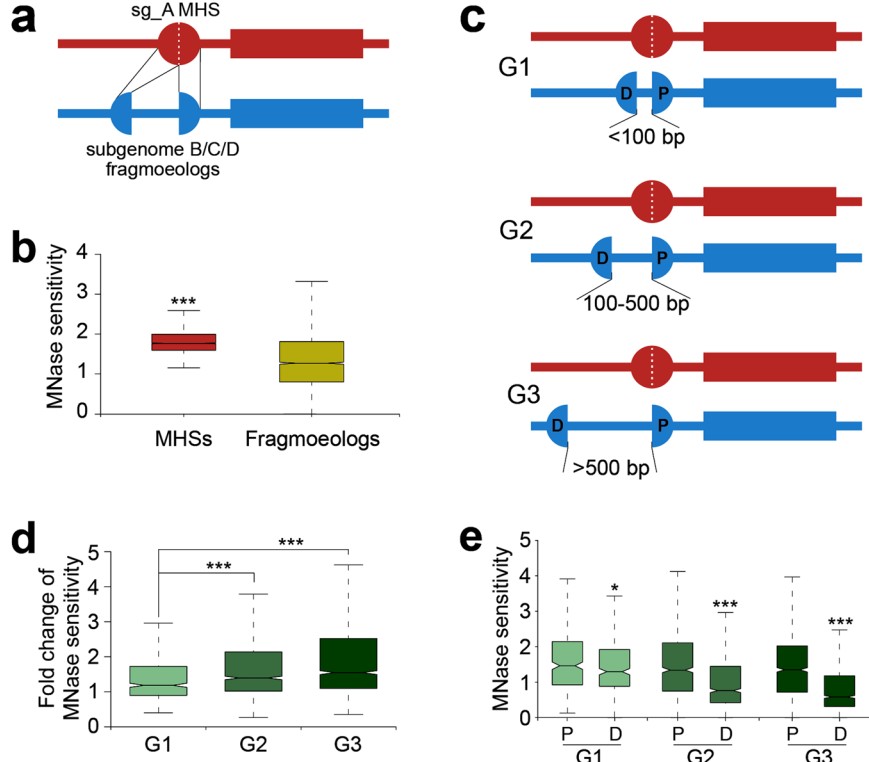

**Fig. 6 | MHS fragmentation and its impact on subgenome dominance. a** A schematic illustration of an MHS in subgenome A and its fragmented homoeologs in subgenomes B/C/D. The rectangles indicate a pair of homoeologous genes associated with the MHS and the fragmented homoeologs. **b** Comparison of MNase sensitivity between subgenome A MHSs (n = 3792) and their fragmented homoeologs from B/C/D subgenomes (n = 7883, p < 2.2e−16). The number of MH-seq reads associated with MHSs or their fragmented homoeologs were used to represent MNase sensitivity. **c** A schematic illustration of two fragmented homoeologs separated by different distances. D: distal homoeolog; P: proximal homoeolog.

**d** Impact of the distance between two fragmented homoeologs on MNase sensitivity change between the MHSs and their fragmented homoeologs (n$_{g1}$ = 511; n$_{g2}$ = 1289; n$_{g3}$ = 1712, G1 vs. G2: p = 1.1e−11; G1 vs. G3: p < 2.2e−16; G2 vs. G3: p = 8e−8). **e** Comparison of the MNase sensitivity levels between proximal and distal homoeologs with different distances (G1: p = 2.5e−2; G2: p < 2.2e−16; G3: p < 2.2e−16). The lower and upper boundaries of each box indicate 25th and 75th percentile, the center line indicates the median, and the whiskers extend to 1.5× IQR in (**b**) and (**e**). ***p < 0.001, two-sided Mann–Whitney U test. Source data are provided as a Source Data file.

(p = 1.8e−5, chi-square test) than the comparison of A/B gene pairs with presence/absence of MHSs.

Collectively, these results showed that more MHS-related sequences were lost in subgenome B, compared to subgenome A, after polyploidization and suggest that the MHS deletions negatively impacted the expression of the cognate genes in subgenome B. In most cases, the absence of corresponding MHS in subgenome B was accompanied by the loss of the cognate subgenome B genes, suggesting that these MHSs are functionally associated with these genes.

**Fragmentation of MHSs and its impact on gene expression**

To further analyze the MHSs associated with subgenome A, we aligned subgenome A MHS sequences to the B/C/D subgenome sequences. In many cases, the aligned subgenome A MHS sequence was not contiguous, but broken into multiple fragments in the B/C/D subgenomes. These results suggest that the cis-regulatory sequences (CRSs) have become fragmented during evolution of the B/C/D subgenomes. We refer to these fragmented MHS regions in B/C/D as "fragmoeologs" thereafter (Fig. 6a). A total of 3792 subgenome A MHSs aligned with multiple fragmoeologs from subgenomes B/C/D (1307 with B; 1239 with C; 1246 with D), respectively. The majority of these subgenome A MHSs (3512/3792, 92.6%) were split into two fragmoeologs, while the remaining were split into greater than two fragmoeologs. We investigated the impact of MHS fragmentation on MNase sensitivity by comparing the MH-seq reads aligned with subgenome A MHS and the corresponding fragmoeologs. The average MNase sensitivity associated with the fragmoeologs was significantly lower (p < 2.2e−16,

Mann–Whitney U test) than their corresponding subgenome A MHSs (Fig. 6b).

We focused on the 3512 subgenome A MHSs split into two fragmoeologs for further analysis. The distance between the two fragmoeologs ranged from 50 to 14,999 bp. We divided the 3512 pairs into three groups based on the distance between the two fragmoeologs, including G1 (<100 bp, 511/3512, 14.6%), G2 (100–500 bp, 1289/3512, 36.7%), and G3 (>500 bp, 1712/3512, 48.7%) (Fig. 6c). The fold change of chromatin accessibility between MHSs and their fragmoeologs was found to be positively correlated with the distance between the two fragmoeologs (Fig. 6d). The two fragmoeologs were referred to as the proximal (P) fragmoeolog and distal (D) fragmoeolog based on their relation to the cognate gene (Fig. 6c). The P and D fragmoeologs showed a relatively similar MNase sensitivity level when they are separated by <100 bp (Fig. 6e). In contrast, the MNase sensitivity levels of the D fragmoeolog became significantly lower than P fragmoeolog as the distance between them increased (Fig. 6e). Thus, the CRS associated with the D fragmoeolog will likely have a reduced regulatory role on its cognate gene with increasing distance after the fragmentation.

We next examined the effect of MHS fragmentation on gene expression. A total of 1859 pairs of MHSs and their fragmoeologs were associated with 1796 homoeologous gene pairs between subgenome A and subgenomes B/C/D. Nearly 30% (535/1796) of the subgenome A genes showed 1.5-times higher expression levels than their homoeologous genes associated with fragmoeologs.

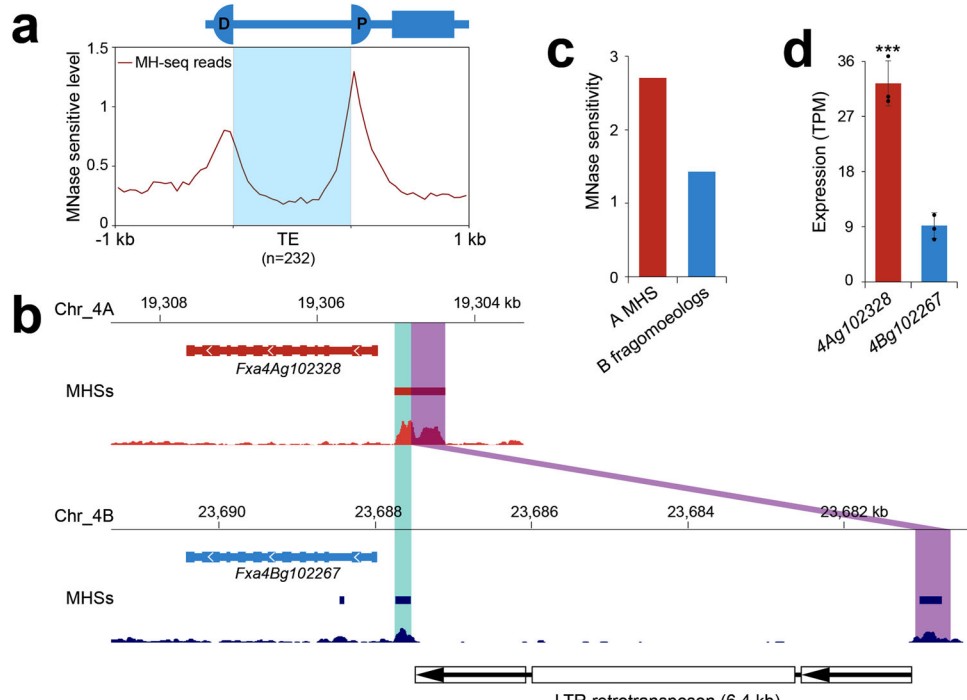

**Fig. 7 | Fragmentation of MHSs and impact on expression of cognate gene.**
**a** MNase sensitivity levels across TEs (n = 232) inserted in MHSs associated with
subgenomes B/C/D. Each TE and its 1 kb flanking regions were divided into 20 bins.
The MNase sensitivity level is represented by the number of MH-seq reads nor-
malized by the length of each bin. **b** Identification of an MHS associated with gene
*Fxa4Ag102328* in subgenome A and its fragmoeologs in subgenome B. The frag-
mentation in subgenome B was caused by insertion of an LTR retrotransposon in
the middle of the MHS. Cyan and purple boxes indicate the MHS in chromosome 4A

and its fragmoeologs in chromosome 4B. **c** Comparison of MNase sensitivity
between the MHS and its fragmented homoeologs. MH-seq reads associated with
the two homoeologs were combined. MH-seq read numbers were normalized by
the sequence length of the MHS or the total length of the two homoeologs.
**d** Comparison of expression levels between *Fxa4Ag102328* and *Fxa4Bg102267*
($p = 3.4e−4$). The data are presented as mean ± s.d. (n = 3 biological replicates).
***$p < 0.001$, one-tailed *t*-test. Source data are provided as a Source Data file.

## Frequencies of MHS fragmentation among different subgenomes

Next, we investigated the relative frequency of MHS fragmentation
occurred in subgenomes A and B. By aligning the sequences of sub-
genome B MHSs with subgenome A sequences, 1208 subgenome B
MHSs were found to be associated with fragmoeologs in subgenome A,
which is similar to the number of subgenome A MHSs (1307) asso-
ciated with fragmoeologs in subgenome B. We then investigated
whether each fragmentation occurred after polyploidization by com-
paring the sequences with the diploid progenitors. For the 1208 sub-
genome B MHSs that became fragmented in subgenome A, 16% (198/
1208) of them aligned with a continuous sequence in the *F. vesca*
genome, indicating that these sequences were likely fragmented after
polyploidization. By contrast, for the 1,307 subgenome A MHSs that
became fragmented in subgenome B, a significantly higher percentage
($p < 2.2e−16$, chi-square test) of them (37%, 481/1307) aligned with a
continuous sequence in the *F. iinumae* genome.

We also identified 1019/1041 subgenome C/D MHSs associated
with fragmoeologs in subgenome A. We investigated if each frag-
mentation occurred after polyploidization using *F. iinumae* as the
diploid comparator for both C and D subgenomes. We found that 21%
(218/1019) of subgenome C MHSs fragmented in subgenome A aligned
with a continuous sequence in the *F. vesca* genome, indicating that
these sequences were likely fragmented after polyploidization. We
identified 1239 subgenome A MHSs associated with fragmoeologs
in subgenome C, a significantly higher percentage ($p < 2.2e−16$, chi-
square test) of them (44%, 549/1239) aligned with a continuous
sequence in the *F. iinumae* genome. Similarly, 21% (221/1041) of
subgenome D MHSs aligned with a continuous sequence in the
*F. vesca* genome, and a significantly higher percentage (44%, 549/1246,

$p < 2.2e−16$, chi-square test) of subgenome A MHSs fragmented in
subgenome D aligned with a continuous sequence in the *F. iinumae*
genome.

Collectively, these results show that more MHSs in subgenome B/
C/D have become fragmented after polyploidization compared to
subgenome A.

## Transposable elements and MHS fragmentation

Given that most insertions between fragmoeologs were larger in size
(Group 3), we hypothesized that the fragmentation of MHSs is likely
caused by insertions of transposable elements (TEs). To test this
hypothesis, we analyzed the 3512 insert sequences between two frag-
moeologs for homology to TEs. We found that 56% (1970/3512) of the
sequences show >50% overlap with known TEs. This association
increased with insert size, as nearly 75% of insert sequences of frag-
moeologs separated by >500 bp were related to TEs.

Previous studies indicated that TEs can serve as regulatory
sequences[69–72]. Therefore, it raised the question about whether some
of the TE sequences were recruited to be part of the MHSs after the
insertion events. Indeed, we found that for ~12% (232/1970) of
fragmoeolog-inserted TE-related sequences, ≥50 bp of the TE
sequences were identified to be part of the MHSs. We mapped all MH-
seq reads associated with these 232 TEs and their 1 kb flanking regions.
The ends of the inserted TE sequences showed higher chromatin
accessibility than the center of the TE sequences (Fig. 7a). These
results indicate that these TE regions were recruited into the pre-existing
open chromatin environment as part of the MHSs, which can poten-
tially impact the transcription of their cognate genes.

We annotated 1970 TE-related sequences between fragmoeologs.
Long terminal repeat (LTR) retrotransposon-related sequences were

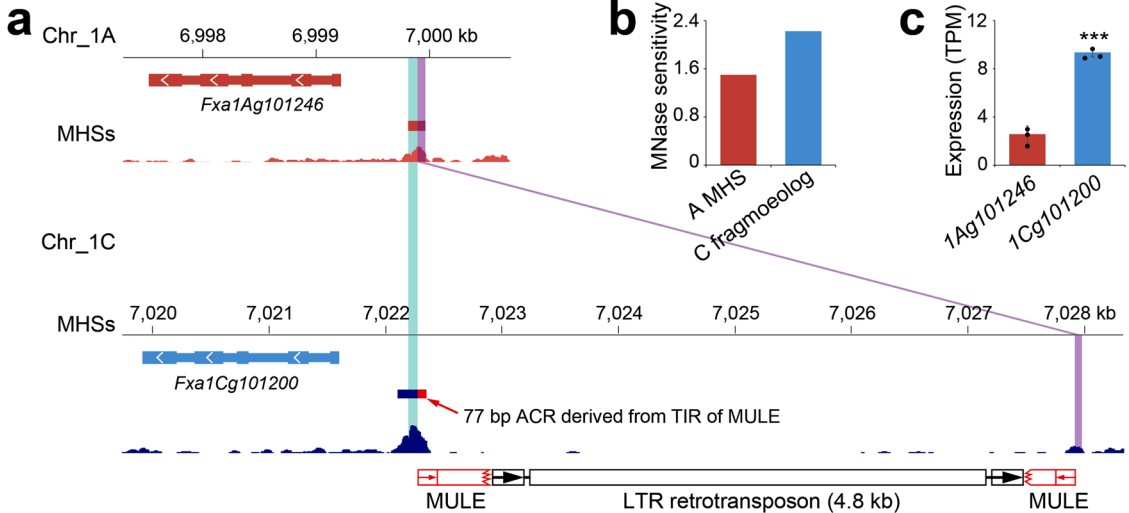

**Fig. 8 | A case of enhanced MNase sensitivity of a fragmented MHS and its impact on gene expression. a** Identification of an MHS associated with gene *Fxa1Ag101246* in subgenome A and its fragmoeologs in subgenome C. The fragmentation in subgenome C was caused by insertions of two nested TEs where a 4.8 kb-LTR retrotransposon (depicted as black boxes and arrows) inserted into an 800-bp MULE element (depicted as red boxes and arrows). Cyan and purple boxes indicate the MHS in chromosome 1A and its fragmoeologs in chromosome 1C. The red rectangle indicates a 77-bp ACR contributed by the TIR of the MULE element. (**b**) Comparison of MNase sensitivity between the MHS in subgenome A and the expanded proximal fragmoeolog in subgenome C. MH-seq read numbers were normalized by the sequence length of the two MHSs. **c** Comparison of the expression levels between *Fxa1Ag101246* and *Fxa1Cg101200* ($p = 6.1e-5$). The data are presented as mean ± s.d. (n = 3 biological replicates). ***$p < 0.001$, one-tailed $t$-test. Source data are provided as a Source Data file.

most abundant, but were not over-represented compared to their average frequencies in the strawberry genome. In fact, *Gypsy*-like retrotransposons appeared to be under-represented (Supplementary Fig. 8). DNA transposons were more variable in representation. DTA (*Ac/Ds/hAT*) and DTH (*PIF/Harbinger/Tourist*) elements were found to be 2- to 3-fold more enriched in the insert sequences, while DTM (*MuDR/Mutator/Mu/MULE*) elements demonstrated a moderate over-representation (~30%). In contrast, DTC (*En/Spm/dSpm*/CACTA) elements were underrepresented (Supplementary Fig. 8). As a result, three DNA transposon superfamilies (out of 11 superfamilies of TEs) were found to be preferentially inserted within the fragmoeologs.

## Cases of TE insertions and their impact on gene expression

Most TE insertions caused reduced chromatin accessibility of the fragmoeologs (Fig. 6b), which may negatively impact the expression of the cognate genes. *Fxa4Ag102328* from subgenome A and *Fxa4Bg102267* from subgenome B are a pair of homoeologous genes, which encode a sulfite oxidase and play a role in sulfite detoxification[73]. The homoeologous copies of this gene were not detected in subgenomes C and D, which were likely lost after polyploidization. We identified a 648-bp MHS in the 5′ of *Fxa4Ag102328* (Fig. 7b). Its homoeologous sequence in subgenome B was broken into two fragmoeologs (457 bp and 216 bp, respectively) in the 5′ of *Fxa4Bg102267* due to the insertion of a 6.4-kb LTR retrotransposon. This retrotransposon includes a pair of approximately 1.5-kb LTRs that contain only 6 bp mismatches (Fig. 7b). The age of this retrotransposon was estimated to be 0.16 MYA, which confirms that the insertion occurred after the final polyploidization event, ~1 MYA[51]. The average chromatin accessibility of the MHS in subgenome A was nearly 2 times higher than its two fragmoeologs in subgenome B (Fig. 7c), while the expression level of *Fxa4Ag102328* is more than three times higher than *Fxa4Bg102267* in the leaf tissue (Fig. 7d).

Interestingly, we found a total of 11 cases in which the fragmoeologs in B/C/D subgenomes showed elevated chromatin accessibility than their corresponding subgenome A MHSs. *Fxa1Ag101246* from subgenome A and *Fxa1cg101200* from subgenome C are a pair of homoeologous genes that encode a Sec14p-like phosphatidylinositol transfer family protein[74]. We identified a 152-bp MHS in the 5′ of

*Fxa1Ag101246* (Fig. 8a). Its homoeologous sequence in subgenome C was broken into two fragmoeologs (96 bp and 63 bp, respectively) in the 5′ of *Fxa1cg101200* due to insertions of two nested TEs where a 4.8-kb LTR retrotransposon inserted into an 800-bp *Mutator-like* element (MULE) (Fig. 8a). The two LTRs of the retrotransposon are identical with the exception of a 9-bp indel, suggesting a very recent insertion. We found that the 63-bp distal fragmoeolog in subgenome C lost the chromatin accessibility. However, the ACR related to the 96-bp proximal fragmoeolog expanded in both 5′ and 3′ directions, including a 77-bp sequence recruited from the terminal inverted repeat (TIR) of the MULE element (Fig. 8a). The MNase sensitivity of this proximal fragmoeolog in subgenome C was 1.5 times higher than the MHS in subgenome A (Fig. 8b). Consequently, the expression level of *Fxa1cg101200* was more than three times higher than that of *Fxa1Ag101246* in leaf tissue (Fig. 8c).

## Discussion

TE density has been previously associated as a driver of subgenome dominance in allopolyploids, including negatively impacting the expression of neighboring genes[28,34]. In this study, we demonstrate that TEs can insert into ACRs and alter the chromatin accessibility level in the regions (Fig. 6b). The newly inserted TEs will likely be inactivated via de novo DNA methylation, which may extend into the flanking ACRs and cause reduced chromatin accessibility. In addition, a distal fragmoeolog can be moved far away from its cognate gene after the TE insertion (Fig. 6c), which may generate a physical barrier for the fragmoeolog to interact with the promoter of its cognate gene. Therefore, TE-mediated ACR fragmentation will generally have a negative impact on the expression of cognate genes.

Interestingly, we observed cases in which TE insertions result in an enhanced MNase sensitivity of the ACRs (Fig. 8). In this case, the sequence exhibiting enhanced MNase sensitivity is within the TIR of a MULE. TIRs of MULEs are known to harbor regulatory sequences[75] and are more often associated with open chromatin than average genomic sequences[35]. As MULEs are one of the families of TEs enriched in insertions within ACRs (Supplementary Fig. 8), they may play a role in the redistribution of CREs and alteration of the expression pattern of genes. In maize, specific regions within the LTRs of retrotransposon

were frequently identified as ACRs[71]. Such regions may be recruited to be part of the ACRs after an insertion (fragmentation) event. Thus, ACR fragmentation can either negatively (majority) (Fig. 7) or positively (minority) impact the expression of cognate genes (Fig. 8). It is important to note that the frequency of ACR fragmentation is likely underestimated in the present study. Fragmentation events may become undetectable due to loss (deletion) of fragmoeologs or multiple rounds of TE insertions, which may create barriers for the sequence alignment-based fragmentation identification pipeline.

Accessible chromatin (or open chromatin) regions, which are hypersensitive to various nucleases, are known to contain CREs[76]. The level of chromatin accessibility of each ACR can be measured by the density of sequence reads from nuclease assays. Chromatin accessibility levels of the 5′ ACRs of active genes are positively correlated with the expression levels of the genes[77]. In addition, enhancer functions of ACRs located outside of promoter regions, including those located in the introns, have been demonstrated in several plant species[25,71,78]. Therefore, the number of ACRs and the chromatin accessibility level of each ACR can be used to measure and compare the activity and divergence of duplicated genes or homoeologous genes. We recently demonstrated that gain or loss of DNA sequences and mutation of *cis*-regulatory elements located within flanking ACRs can change the balance of the expression level and/or tissue specificity of duplicated genes in soybean[25]. In the current study, we found that the dominant subgenome A contains a greater number of total MHSs and MHS per gene than the submissive B/C/D subgenomes. Subgenome A also suffered fewer losses of MHS-related DNA sequences and fewer MHS fragmentations. Thus, the gene expression dominance of subgenome A is well correlated with the quantities of cognate ACRs (Fig. 1) and chromatin accessibility level of the ACRs (Fig. 2) associated with the genes.

Genome sequencing and phylogenetic analyses based on transcriptomes of every described diploid *Fragaria* species, together with the geographic distributions, natural history, and genomic footprints of the diploid species, suggested that the octoploid strawberry originated from fusion of a hexaploidy species and *F. vesca* approximately 1.1 MYA. More specifically, *F. vesca* subsp. *bracteata* was proposed to be the most likely diploid progenitor of the dominant A subgenome and maternal parent[39,44]. Since *F. vesca* subsp. *bracteata* is endemic to the western part of North America, this species should well adapt to the environment in North America before it was fused to a hexaploid strawberry – whose diploid progenitors are estimated to be from the 'Old World'. Thus, the locally adaptive genes in *F. vesca* are possibly important for the survival and diversification of the newly formed octoploid strawberry across North and South America. This hypothesis is supported by the abundant stress-responsive genes enriched in the collection of subgenome A-specific genes (Fig. 4). Similarly, previous studies of the octoploid strawberry genome revealed enrichment of disease resistance genes towards the dominant subgenome[39,79]. Response to pathogen infection was also reported to be associated with the subgenome dominance in *Brassica napus*[80]. In addition, selective sweeps associated with domestication of octoploid strawberry were shown to be significantly biased towards (*p*-value < 0.001) the dominant A subgenome[46]. Here, we show that subgenome A MHSs have undergone fewer nucleotide mutations, sequence losses, and fragmentations compared to subgenome B/C/D MHSs (Figs. 3, 5, and 6). These results suggest that subgenome A has likely undergone stronger selective constraint on its CRSs after polyploidization compared to the other subgenomes. Therefore, the environmentally adaptive advantage of *F. vesca*, differences in transposable element content between subgenomes and selective maintenance of CRSs may have played an important role in establishing the dominance of subgenome A in octoploid strawberry.

Finally, it is worth noting that we did not investigate the impact of hybridization order as part of this study, but it would be worth modeling as part of a future follow-up study.

## Methods

### Preparation of MH-seq libraries and RNA-seq libraries
Three biological replicates of young trifoliate leaves were collected from seedlings grown in greenhouse and growth chamber, respectively. Each collected sample was ground into fine powder using liquid nitrogen and divided into two parts. One was used for construction of MH-seq library and another one was used for construction of RNA-seq library. We conducted RNA-seq from all three biological replicates, but conducted MH-seq using only two of the three biological replicates. For the MH-seq library, nuclei were firstly isolated from 2 g powder following published protocols[81]. Isolated nuclei were suspended in 1.5 ml MNase digestion buffer (MNB, 10% sucrose, 50 mM Tris-HCl, pH 7.5, 4 mM MgCl2, and 1 mM CaCl2) and divided into five 1.5-ml Eppendorf tubes (300 μl per tube). The aliquoted nuclei were digested at 37 °C for 10 min using 0.2 U of MNase (N3755-50UN, Sigma). After MNase digestion, DNA was extracted by adding 300 μl CTAB and incubated at 65 °C for 15 min following the CTAB method[82]. The extracted DNA was detected by running 2% agarose gel in 1x TAE buffer. DNA fragments <100 bp were extracted from gel to prepare MH-seq library following standard Illumina library preparation procedures. MH-seq libraries were sequenced with single-end, 100-bp reads on an Illumina NovaSeq 6000 system by Michigan State University Research Technology Support Facility Genomics core. For the construction of RNA-seq library, total RNA was isolated with KingFisher Pure RNA Plant Kit (Thermo Fisher). RNA libraries were prepared with the KAPA mRNA HyperPrep Kit protocol (KAPA Biosystems). The samples were sequenced with paired-end, 150-bp reads on an Illumina NovaSeq 6000 system by Michigan State University Research Technology Support Facility Genomics core.

### Identification of MHSs and gene expression
Adapters were trimmed out from sequencing reads with trimmomatic v0.36[83] using the parameters "ILLUMINACLIP:/TruSeq3-SE.fa:2:30:10 LEADING:3 TRAILING:3 SLIDINGWINDOW:4:15 MINLEN:30". The clean MH-seq reads were then mapped to Royal Royce reference genome using BWA (0.7.15-r1140) with default parameters[84]. All uniquely mapped reads were used for identification of MHSs by F-seq with a 200-bp bandwidth[56]. To estimate the false discovery rate (FDR) of identified MHSs, we calculated the ratio of number of MHSs identified based on genomic sequencing datasets of Camarosa[39] to the number of MHSs from the MH-seq data. The threshold was set in F-seq to control the FDR < 0.01. MHSs with a length ≥50 bp were retained for further analysis. RNA-seq reads were mapped to strawberry Royal Royce reference genome using HISAT2[85] (version 2.0.0-beta). The TPM (Transcripts Per Million) value, calculated by StringTie[85] (v1.3.3b), was used to represent the expression level of genes from four subgenomes. The average TPM value of each gene was calculated from the three biological replicates.

### Identification of homoeologous genes
Sequences of gene coding regions were aligned between every two of the four subgenomes using BLAST[86] with E-value < 1E−10. The blast results were used to identify syntenic genes between any two subgenomes by MCScanX with default parameters[87]. The identified syntenic genes were determined as homoeologous genes between two subgenomes. The expression levels of a pair of homoeologous genes are considered to be significantly different if the expression level of one copy is at least 1.5 times higher than the other copy. The sequences of MHSs from subgenome A were aligned with each of the genomic sequences of B, C, and D subgenomes using BLAST[86]. MCScanX was used to identify syntenic regions between any two subgenomes with

default parameters according to the blast results[87]. The syntenic regions of the MHSs in B, C, and D subgenomes were determined as homoeologous regions. In addition, some MHSs have matching sequences in B, C, and D subgenomes, but they are not determined as syntenic regions because of rearrangement in the genomic sequences. If these matching sequences whose cognate genes are still homoeologous to the cognate genes of their corresponding MHSs, they are still considered homoeologous regions. The MEME software was used to identify enriched DNA sequence motifs within the subgenome A-specific MHSs with the parameters "-nmotifs 30 -minw 6 -maxw 7 -revcomp".

### Identification of MHSs that lost their homoeologs

Subgenome A genes that lack a syntenic homoeolog in B, C, and D subgenomes are classified as subgenome A-specific genes. TBtools[88] was used to perform GO analysis of the subgenome A-specific genes and the results were displayed by an online tool (www.bioinformatics.com.cn). The subgenome A MHSs that lack matching sequences or whose matching sequences are not linked to the corresponding homoeologous genes in the B, C, and D subgenomes are classified as subgenome A-specific MHSs. Using the same criteria, we identified B-, C-, and D-specific genes and MHSs. To detect that the loss of homoeologous regions happened before or after polyploidization, the syntenic regions of MHSs were detected in *F. vesca* or *F. iinumae* using BLAST and MCScanX. If an MHS from subgenome A does not have a syntenic sequence in subgenome B but has a syntenic sequence in *F. iinumae*, we considered the loss of this homoeologous region in subgenome B occurred after polyploidization. If no syntenic sequence was identified in *F. iinumae*, we considered the loss of this homoeologous region occurred before polyploidization.

### Identification of SNPs/INDELs between hi-MHSs and their homoeologs

The numbers of MH-seq reads associated with each MHS and its homoeolog were calculated, respectively. These numbers were imported into Bioconductor package edgeR[89] to identify the MHS from subgenome A that exhibits a significantly higher level ($P < 0.01$ and logCPM>0) of MNase sensitivity than its homoeolog (hi-MHSs) or a significantly lower level ($P < 0.01$ and logCPM > 0) of MNase sensitivity than its homoeolog (lo-MHSs). To identify SNPs/INDELs between the sequences of MHSs and their homoeologs, we first create the pair alignment of nucleotide sequences from MHSs and their homoeologs using muscle[90]. Then, the SNPs/INDELs were called from the pair alignment of sequences using msa2snp.py (https://github.com/pinbo/msa2snp). To detect the genotypes of SNP/INDELs in the progenitor of a subgenome, we first identified the syntenic regions of the hi-MHS in *F. vesca* and *F. iinumae* through BLAST and syntenic analysis. Then, we detected SNP/INDELs between hi-MHSs and their syntenic regions in the progenitors using muscle and msa2snp.py.

### Annotation and identification of transposable elements

The transposable elements in the Royal Royce reference genome were first identified with EDTA[91]. Since EDTA is not robust with the detection of long interspersed nuclear elements (LINEs)[91], protein sequences related to known LINEs were retrieved from Repbase[92] and used to search against the Royal Royce reference genome (BLASTN, E-value = 1e$^{-5}$). The redundancy of sequences related to LINE proteins was reduced based on the definition of family proposed previously[93]. Thereafter, the 13 most abundant LINE families were manually curated[94], and the remainder of them were considered as LINE fragments. In addition, 18 other TEs that have interrupted MHS (see below), but were completely or partly absent from the EDTA library, were manually curated. The manually curated TE sequences were used to mask EDTA library and the masked portions were excluded to generate a "clean" EDTA library. The "clean" EDTA library, curated TE sequences, and LINE fragments were combined to form the final TE library for the annotation of TEs using EDTA 2.1.1[91]. PanEDTA.sh[95] was used to employ a pangenome approach, incorporating other available strawberry genomes to reduce the quantity of false positives and improve annotation consistency.

### Identification of fragmoeologs

When an MHS aligns with more than two homoeologs and the distance between the homoeologs ranges from 50 bp to 15 kb, the multiple homoeologs of this MHS were fragmented homoeologs (fragmoeologs). The number of MH-seq reads that mapped to MHSs or to the fragmoeologs were calculated to compare chromatin accessibility between the MHSs and the fragmoeologs. The insertion sequences (between fragmoeologs) that show >50% overlap with annotated TEs were considered as TE-derived insertions. To detect the fragmented homoeologs occurred in subgenome B after polyploidization, we aligned the sequence of MHS from subgenome A with the genome of *F. iinumae*. If the syntenic region of MHS is intact in *F. iinumae*, the fragmented homoeologs were considered as generating after polyploidization. Otherwise, they were generated before polyploidization.

### Reporting summary

Further information on research design is available in the Nature Portfolio Reporting Summary linked to this article.

## Data availability

MH-seq datasets and RNA-seq data have been submitted to NCBI under the BioProject accession PRJNA972699. The genomic sequencing data of strawberry cultivar Camarosa was downloaded from NCBI Sequence Read Archive under SRR8358386 and SRR8358387. Source data are provided with this paper.

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

## Acknowledgements

This research is supported by the National Science Foundation grant IOS-1740874 to N.J. and ISO-2029959 to C.N., J.J. and P.P.E.

## Author contributions

J.J. and C.F. designed the experiments. C.F. conducted the experiments. C.F., N.J., S.J.T., A.E.P., G.A., C.N., P.P.E. and J.J. analyzed the data. C.F., P.P.E. and J.J. wrote the manuscript.

## Competing interests

The authors declare no competing interests.
