## [Peer Review File · Nature Communications]

Dynamics of accessible chromatin regions and subgenome dominance in octoploid strawberryReviewers' Comments:

Reviewer #1:

Remarks to the Author:

The authors identify accessible chromatin regions (ACRs) in allooctoploid strawberry leaf using MNase-hypersensitivity sequencing. They discuss the distribution of these sites throughout the subgenomes with an attempt to discuss genome dominance in polyploids.

There are two related major problems with the paper as written. First, there is no discussion of the much disputed subgenome assignments used in this paper. <https://www.nature.com/articles/s41588-019-0543-3> , <https://academic.oup.com/mbe/article/38/2/478/5907923> , <https://www.nature.com/articles/s41467-023-38560-z> , and <https://www.nature.com/articles/s41477-023-01473-2> . The difference in subgenome assignments is not trivial given the nature of the paper, the authors argue that the "A" subgenome is exceptional in its statistical significance compared to the other subgenomes. In Figure 1 the D subgenome appears to be higher in the average number of MHSs per gene, albeit not significantly as measured by ANOVA. But if subgenomes were called correctly, this may be a significant difference that is not discussed by the authors. There are other such comparisons in the later Figure panels as well.

Second, and more importantly, there is no discussion about the impact of the order of hybridization of the allooctoploid on the observed "subgenome dominance". As discussed in relation to allohexaploid *Camelina* (<https://onlinelibrary.wiley.com/doi/full/10.1111/tpj.15931>) and both *Camelina* and Strawberry (<https://www.nature.com/articles/s41467-023-38560-z>) for allopolyploids with more than 2 subgenomes, we must consider the order of hybridization when comparing "dominant" subgenomes to allotetraploids. Specifically, the "A" subgenome discussed in this paper was the final to be added to the growing allooctoploid, therefore it has experienced far less relaxed selection, and we expect it to have the highest expression and longest MHSs just based on its evolutionary history. The observed results in this paper are all consistent with that, and may indeed have nothing to do with subgenome dominance at all. A large scale rewrite would be necessary to address this as the true null hypothesis of many of the experiments is that we expect the "A" subgenome to appear dominant in all these measurements. Therefore the results are not significant evidence of subgenome dominance. A correct argument for dominance would need to relate the difference in MNase activity given the time of the subgenome in the polyploid context indicating relaxed selection.

Additionally, related to both points above, there is no discussion that the "B" subgenome (closer to *F. iinumae*) was added as the 3rd subgenome to the ancestral allohexaploid, yet often has fewer/shorter MHSs compared to the two with a longer time in a polyploid context under relaxed selection. These may be reflective of a true dominance being measured, especially if the "C/D" subgenomes were called correctly.

Reviewer #2:

Remarks to the Author:

The present manuscript conducted a genome-wide accessible chromatin region (ACR) analyses in cultivated octoploid strawberry with A, B, C, D subgenomes, and performed correlation analyses among ACRs and homoeologs, gene expression, transposable elements. This study provided information to explain the mechanism for A subgenome dominance in octoploid strawberry. The followings are suggested to be considered in revising the manuscript.

(1) The authors indicated the 'cis-regulatory' roles of MHSs in Title, Abstract and Introduction, but this word didn't appear in Result and Discussion. Chromatin accessibility reflected by MHS should be the topic of the whole study.

(2) The authors mainly compared the MHS, gene expression, homoeologs, fragmoeologs and TEs in A and B subgenomes (e.g. 'Loss of MHS-related sequences after polyploidization', 'Frequencies of MHS

fragmentation associated with subgenomes A and B', etc.), and the comparisons between subgenome A and C, D were rarely described in the manuscript. The later comparisons were important to understand and confirm the molecular regulatory mechanism for A subgenome dominance.

(3) In Discussion part, the relationship between chromatin accessibility and A subgenome dominance should be thoroughly discussed.

(4) Line 480-481 showed 'The average TPM value of each gene was calculated from the three biological replicates', but Line 448 showed 'Two biological replicates of young trifoliolate leaves were collected from seedlings'. Please check the number of biological replicates.

(5) Line 266, '1.5-times lower expression levels', the '1.5-times' and 'lower' seems contradictory.

(6) Line 72, '(Fragaria x ananassa)', the 'x' should be '×'.

(7) Line 78, 'Fragaria vesca ssp. bracteata', the 'ssp.' shouldn't italic.

(8) Line 427-428, 'Bracheata' should be 'bracheata'.

Reviewer #3:

Remarks to the Author:

This manuscript entitled "Divergence and fragmentation of cis-regulatory sequences and subgenome dominance in octoploid strawberry" by Fang et al. generated a genome-wide map of accessible chromatin regions (ACRs) in cultivated strawberry using MHS sequencing, and compared them within the four subgenomes A, B, C, and D. Their data demonstrated the biased evolution of these MHS sites. I have some concerns and questions.

1. If I understand correctly, the authors identified syntenic MHS sites or genes by only comparing them within the four subgenomes. I think this is not strict. Based on my experience, the authors should identify a closely relative outgroup to examine whether these homologous MHS sites or genes have orthologs or syntelogs in the outgroup. For example, sorghum has been used as the outgroup for comparing the two subgenomes in maize. If the syntenic MHS sites or genes are identified within the same genome, some of the homologous pairs may be generated by segmental duplication or other duplication events. Not all of them are generated by polyploidization.

2. The description of the paper is not very clear. For example, how many genes and MHS sites in subgenomes A, B, C, and D? How many of these genes and MHS sites have homoeologous pairs in the genome, and how many of them are singletons? Please describe this kind of information at the beginning of your results.

3. Results, page 5, lines 133-135, "For each homoeologous gene pair, one copy was defined to have an elevated expression if it expressed at least 1.5-fold higher than the other copy.". Normally people use 2-fold higher to tell the expression dominance. Why did the authors use a lower standard (1.5-fold higher) here?

4. Results, page 6, line 144, for the 6,824 dominantly expressed genes from subgenome A, how many of them have chromatin accessibility data?

5. Figure 2d and 2e, the colors for subgenomes A and C are too close.

6. Results, page 7, lines 175-178, this part is confusing. When the authors talked about 2,034 pairs of control-MHSs, did the authors mean the two homoeologous regions show similarity and both are MHSs? I guess my question for the whole paper is when the authors tried to identify MHS pairs, did they only consider the sequence homology or in addition to the sequence similarity, both regions are MHS sites (ACRs)? I am asking this question because we detected many ACRs in one subgenome have homoeologous sequences in the duplicated subgenome, but lost chromatin accessibility, which means one is an ACR, its homoeologous is not.

7. Results, page 7, lines 179-181, "The numbers of SNPs/INDELs between hi-MHSs and their homoeologs were significantly greater than those between control-MHSs and their homoeologs (Figure 3d)." This data is kind of surprising. I expected that hi-MHSs are more conserved and should have fewer SNPs/INDELs. How about the SNPs/INDELs between hi-MHSs and lo-MHSs?

8. Figure 3e, it's hard to understand. Please add more details in the figure legend.

9. Results, page 8, lines 216-217, "These genes were referred to as subgenome A-specific genes.". Do these specific genes have syntelogs in closely related species? My main question is how confident they are all genes.

10. Results, page 10, lines 293-294, "We focused on the 3,512 subgenome A MHSs split into two fragmoeologs for further analysis." How many of these fragmoeologs of the 3,512 subgenome A MHSs are also MHSs?

11. Results, page 11, lines 297-304. The authors wanted to show the point that the fold change of chromatin accessibility between MHSs and their fragmoeologs was positively correlated with the distance between the two fragmoeologs. I guess another important factor is how big the remaining part near the gene side matching the MHSs. I anticipate that the fold change of chromatin accessibility between MHSs and their fragmoeologs is negatively correlated with the length of the retained part of the fragmoeologs. For example, if one fragmoeolog has 90% fragment still matching the homoeologous MHS, the fragmoeolog may still have high chromatin accessibility because some motifs at the fragmoeolog are not disrupted by the insertion.

12. Results, page 12, section "Transposable elements and MHS fragmentation" and Figure 7a. Are these TEs orientated? If these TEs are orientated into proximal and distal ends depending on the flanking MHSs, do the proximal ends show higher MNase sensitive level?

13. Results, page 5, line 134, change "if it expressed" into "if it was expressed".

14. Results, page 9, line 246, change "sequence" into "sequences".

Response to comments from Reviewer #1

The authors identify accessible chromatin regions (ACRs) in allooctroploid strawberry leaf using MNase-hypersensitivity sequencing. They discuss the distribution of these sites throughout the subgenomes with an attempt to discuss genome dominance in polyploids.

There are two related major problems with the paper as written. First, there is no discussion of the much disputed subgenome assignments used in this paper.

<https://www.nature.com/articles/s41588-019-0543-3>,

<https://academic.oup.com/mbe/article/38/2/478/5907923>,

<https://www.nature.com/articles/s41467-023-38560-z>, and

<https://www.nature.com/articles/s41477-023-01473-2>. The difference in subgenome assignments is not trivial given the nature of the paper, the authors argue that the "A" subgenome is exceptional in its statistical significance compared to the other subgenomes. In Figure 1 the D subgenome appears to be higher in the average number of MHSs per gene, albeit not significantly as measured by ANOVA. But if subgenomes were called correctly, this may be a significant difference that is not discussed by the authors. There are other such comparisons in the later Figure panels as well.

Response: Thank you for this comment. We agree, due to the complexity of the composition of the C and D subgenomes, in part due to the evolutionary history of homoeologous exchanges, but also due to differences in methodological approaches, there is an ongoing debate across the strawberry community regarding the assignment of individual chromosomes to each subgenome. The discussion certainly goes beyond the papers that are listed above, and is partly reviewed previously (Hardigan et al., 2021).

As you correctly point out, depending on the question being asked, the chromosomes assignments to each subgenomes could impact the results and conclusions. Fortunately, the chromosomes assigned to the A subgenome are consistent across all previous studies including the ones you listed above. There is disagreement for only one of the chromosomes assigned to the B subgenome based on only a single previous study (Sargent et al., 2016). The majority of disagreements are for some of the chromosomes assigned to the C and D subgenomes – only chromosomes 1, 4 and 7 are in common among recent studies.

To specifically address your concern “In Figure 1 the D subgenome appears to be higher in the average number of MHSs per gene, albeit not significantly as measured by ANOVA. But if subgenomes were called correctly, this may be a significant difference that is not discussed by the authors”, we have examined the possible results if we randomly assign chromosomes 2, 3, 5, and 6 to the C or D subgenomes, with a total of 15 possible combinations (**Figure 1**). Taking one combination as an example, we re-assigned chromosome 2C as chromosome 2D and chromosome 2D as chromosome 2C (**Figure 1, b**). We re-calculated and compared the average MHS number across subgenome A, B, and the newly created subgenome C and D. After performing the analysis for all 15 combinations, we found that the average MHS number of subgenome D is still not significantly higher than subgenome B or C (**Figure 1**), but is significantly lower than subgenome A. Similarly, we also examined the average MHS length per

gene and the average number of reads per gene among the four subgenomes in the same 15 combinations. We obtained similar results that the numbers from subgenome D are not significantly higher than those from subgenome B or C, but are significantly lower than subgenome A.

Figure 1. Average number of MHS per gene in each subgenome. In (a), the 7 chromosomes from subgenome C and the 7 chromosomes from subgenome D were assigned based on the original report of Royal Royce reference genome. Purple boxes indicate the chromosomes from subgenome C and cyan boxes indicate chromosomes from subgenome D. In (b-p), we exchanged the chromosome 2, 3, 5, 6 between subgenome C and D in a total of 15 combinations. The average number of MHSs per gene was compared among the four subgenomes in every combination. The total numbers of MHSs were normalized in the four subgenomes by dividing to their total number of genes from each subgenome. Means that do not share a letter are significantly different ($p < 0.01$, one-way ANOVA with Games-Howell *post-hoc* test).

Furthermore, our analyses in this manuscript are focused on comparing the dominant A subgenome to the three submissive subgenomes (B, C and D). Thus, we address your concern (which chromosomes are assigned to subgenome C vs D) by comparing the dominant A to the submissive B subgenomes, or A compared to C and D subgenomes combined, and the dominant A subgenome against all three B/C/D submissive subgenomes combined. Thus, if there are any chromosome mis-assignment, we would only be swapping chromosomes between two submissive subgenomes that are already combined as part of a group, which are then compared to the A subgenome. When we grouped subgenome C and D together and calculated the average number of MHS per gene by dividing the total number of MHSs from subgenome C and D to the total number of genes in these two subgenomes. We observed that the numbers of MHSs per gene are comparable between subgenome B and the combined subgenomes C/D, and both are significantly lower than that of subgenome A (Figure 2a). Similar results were also obtained when subgenomes B, C, and D were combined (Figure 2b). We have added this analysis to the revised manuscript.

Figure 2. Comparison of chromatin accessibility between subgenome A and combined subgenomes B/C/D. **(a)** The total number of MHSs (left), MHS length (middle), and MH-seq read number (right) were combined from subgenomes C and D. The combined values were normalized by dividing to the total number of genes from subgenomes C/D. **(b)** The total number of MHSs (left), MHS length (middle), and MH-seq read number (right) were combined from subgenomes B/C/D. The combined values were normalized by dividing to the total number of genes from subgenomes B, C, and D. Means that do not share a letter are significantly different ($p < 0.01$, one-way ANOVA with Games-Howell *post-hoc* test). *** $p < 0.001$, Mann-Whitney *U* test.

We also re-ran the analysis shown in Figure 1C and D by comparing the seven homoeologous chromosome sets separately, thus removing any potential subgenome assignment issues. The overall results were consistent with our previous analysis for all chromosomes except chromosome 7, which is not one of the chromosomes in question -- please see **Figure 3** below. We have included these results in the revised manuscript.

Figure 3. Comparisons of chromatin accessibility among the four chromosomes in each of the seven homoeologous groups. **(a)** Average number of MHSs per gene of every chromosome in each homoeologous group. **(b)** Average MHS length per gene of every chromosome in each homoeologous group. **(c)** Average number of MH-seq reads per gene of every chromosome in each homoeologous group. Reads of all MHSs associated with the same gene were used for calculation. The total number of MHSs, MHS length, and MH-seq read number were normalized by dividing to their total number of genes from each chromosome. Means that do not share a letter are significantly different ($p < 0.01$, one-way ANOVA with Games-Howell post-hoc test).

We have added the following two paragraphs in the Introduction of the revised manuscript for a more accurate description of the current status of chromosome assignments in strawberry:

While there is community consensus about which chromosomes are assigned to the A subgenome, there is some inconsistencies on which chromosomes should be assigned to the B/C/D subgenomes (Liston et al., 2020; Feng et al., 2021; Hardigan et al., 2021; Jin et al., 2023; Session and Rokhsar, 2023). There is strong and consistent support for six of seven chromosomes for the B subgenome, only exception being chromosome 6 from a single study (Sargent et al., 2016). For chromosome assignments to the C and D subgenomes, there is consistent support for only chromosomes 1, 4 and 7 (Tenessen et al., 2014; Sargent et al., 2016; Edger et al., 2019; Session and Rokhsar, 2023). The incongruence of chromosome assignments from these studies stems in part due to differences in their methodological approaches, but also due to a deep history of homoeologous exchanges across the octoploid strawberry genome that have resulted in the gradual blending of the homoeologous chromosomes (Tenessen et al., 2014; Edger et al., 2019; Jin et al., 2023). Thus, none of the chromosomes in the present day octoploid strawberry genome are pure of their diploid ancestry.

Homoeologous exchanges are the reciprocal exchange of DNA during meiosis between homoeologous chromosomes, which were contributed by different parental species but are related by ancestry (e.g., chromosome 1 from both parental species) (reviewed by Deb et al., 2023). Gene conversion events, which results in smaller non-reciprocal conversion events, are also well documented in allopolyploids (Guo et al., 2014; Zhang et al., 2020). The progression towards higher polyploids (e.g., 8x) requires intermediate polyploids, either 4x followed by 6x or the hybridization of two 4x, with each event experiencing their own genomic blending over time and is compounded following multiple polyploidization events. Furthermore, the dominant subgenome preferentially replace regions from the submissive subgenome ranging at rates ranging between 7.3-10.4x (Edger et al., 2019). Collectively, this has made it difficult to assign chromosomes to certain subgenomes, and given the amount of genomic blending, particularly for subgenomes C and D, the features used to guide chromosome assignments to subgenomes often represent a minor fraction of the entire genome. Nevertheless, potential issues associated with chromosome (mis)assignments between submissive subgenomes can be addressed by combining and averaging across the submissive subgenomes (Edger et al., 2017; Edger et al., 2019).

Second, and more importantly, there is no discussion about the impact of the order of hybridization of the allooctoploid on the observed "subgenome dominance". As discussed in relation to allohexaploid *Camelina* (<https://onlinelibrary.wiley.com/doi/full/10.1111/tpj.15931>) and both *Camelina* and *Strawberry* (<https://www.nature.com/articles/s41467-023-38560-z>) for allopolyploids with more than 2 subgenomes, we must consider the order of hybridization when comparing "dominant" subgenomes to allotetraploids. Specifically, the "A" subgenome discussed

in this paper was the final to be added to the growing allooctoploid, therefore it has experienced far less relaxed selection, and we expect it to have the highest expression and longest MHSs just based on its evolutionary history. The observed results in this paper are all consistent with that, and may indeed have nothing to do with subgenome dominance at all. A large scale rewrite would be necessary to address this as the true null hypothesis of many of the experiments is that we expect the "A" subgenome to appear dominant in all these measurements. Therefore the results are not significant evidence of subgenome dominance. A correct argument for dominance would need to relate the difference in MNase activity given the time of the subgenome in the polyploid context indicating relaxed selection.

Response: Yes, hybridization order has been hypothesized to play a potential role in observed divergence patterns among homoeologs and subgenome dominance in higher polyploids. For example, Tang et al. (2012) (Tang et al., 2012), may have been the first, if not among the first, to propose that as an explanation for the presence of a dominant subgenome in Brassica following a whole genome triplication event (paleo-hexaploidy). However, there have been several more recent studies showing that the hybridization order does not dictate which subgenome is dominant. In other words, the last subgenome to be added to the nucleus in an allopolyploid will not necessarily be the most dominant.

For example, a recent study from the Solanaceae (paleo-hexaploid event) showed that the last arriving subgenome has the fewest genes – characteristic of submissive subgenomes (McRae et al., 2022). Similarly, our previous analyses of the hexaploid monkeyflower (*Mimulus peregrinus*) also revealed that the last arriving subgenome (diploid parent) was the lower expressed (submissive) subgenome in both naturally established and resynthesized polyploids (Edger et al., 2017). Both subgenomes of the tetraploid species, whose genome already exhibited fractionation (gene loss), exhibited higher gene expression compared to the diploid parental species in the hexaploid. Our analyses of the monkeyflower genome, as well as octoploid strawberry, mirrors earlier work by Hollister and Gaut (2009), suggesting that certain genomic features, including differences in methylated transposable element content near genes, are often a predictor of which homoeolog is more dominantly expressed. We, as well as others in polyploid community, are certainly not arguing that hybridization order, particularly the time between polyploidization steps, has the potential to impact divergence patterns between homoeologs. However, there are an increasing number of studies showing that hybridization order is not the sole determinant, or a predictor, of which subgenome will become dominant. On the contrary, there are several studies in resynthesized hybrids and polyploids, including one in strawberry that we plan to submit in a few months, that uses different parents, each with genomes and unique transposable element content and DNA methylation landscapes, that match the predictions originally proposed by Hollister and Gaut (2009), later studied in other paleo-polyploids (e.g., Woodhouse et al., 2014), and outlined in various review papers including by us (Bird et al., 2018). For example, we have a set of resynthesized strawberry hybrids that instantly, in the first generation, exhibit expression dominance patterns matching *in silicio* predictions. Nonetheless, there remains a lot of unanswered and exciting research questions that will lead to a more holistic model to explain dominance patterns. In this manuscript, we are investigating how changes in cis regulatory elements may have further contributed to expression divergence of homoeologous genes in the naturally established octoploid strawberry.

Additionally, related to both points above, there is no discussion that the "B" subgenome (closer to *F. iinumae*) was added as the 3rd subgenome to the ancestral allohexaploid, yet often has fewer/shorter MHSs compared to the two with a longer time in a polyploid context under relaxed selection. These may be reflective of a true dominance being measured, especially if the "C/D" subgenomes were called correctly.

Response: The hybridization order, if B subgenome was added third, has not been established for octoploid strawberry. To the best of our knowledge, this has never been modeled/tested in any robust or systematic way. We are certainly not arguing against this as a possibility, but it remains possible that the B subgenome was added in the tetraploid ancestor. We agree that this remains an interesting, and important, research direction to understand the origin of the octoploid strawberry but is beyond the scope of the present study.

Response to comments from Reviewer #2

The present manuscript conducted a genome-wide accessible chromatin region (ACR) analyses in cultivated octoploid strawberry with A, B, C, D subgenomes, and performed correlation analyses among ACRs and homoeologs, gene expression, transposable elements. This study provided information to explain the mechanism for A subgenome dominance in octoploid strawberry. The followings are suggested to be considered in revising the manuscript.

(1) The authors indicated the ‘cis-regulatory’ roles of MHSs in Title, Abstract and Introduction, but this word didn’t appear in Result and Discussion. Chromatin accessibility reflected by MHS should be the topic of the whole study.

Response: We have changed our title as “Birth, death, and mutational dynamics of accessible chromatin regions and subgenome dominance in octoploid strawberry”. We have modified the relevant text in Abstract and Introduction to redirect our focus on “DNA sequences in accessible chromatin regions”.

(2) The authors mainly compared the MHS, gene expression, homoeologs, fragmoeologs and TEs in A and B subgenomes (e.g. ‘Loss of MHS-related sequences after polyploidization’, ‘Frequencies of MHS fragmentation associated with subgenomes A and B’, etc.), and the comparisons between subgenome A and C, D were rarely described in the manuscript. The later comparisons were important to understand and confirm the molecular regulatory mechanism for A subgenome dominance.

Response: We chose to focus A vs. B comparison because the ancestral origin of subgenomes C and D have been a subject of controversy, thus, it is difficult to choose a diploid species as the ancestral genomes for C and D. To respond to the Reviewer’s concern, we have added analyses of comparisons between subgenome A and C/D, by considering *F. iinumae* as the ancestor of

both C and D, which is justified by a recent report on a close phylogenetic relationship between *F. iinumae* and the C/D subgenomes (Jin et al. 2023, Nat. Plants 9: 1252-66).

In the section “Divergence of the DNA sequences associated with MHSs” in Results, we identified syntenic regions in *F. vesca* and *F. iinumae* for a total of 1,742/1,697 pairs of hi-MHSs from subgenome A and homoeologs from subgenomes C/D. We randomly selected the same numbers of pairs of control-MHSs (syntenic subgenome A/C or A/D MHSs without differential MNase sensitivity). We found similar results that the inherited variants from the progenitor species, rather than mutations after polyploidization, represent the major contribution to sequence divergence between subgenome A hi-MHSs and their subgenome C/D homoeologs in octoploid strawberry (Figure 4).

Figure 4. Sequence variation and MHS divergence between subgenome A and subgenomes C/D. **(a)** The number of inherited (left panel) and mutated (right panel) types of SNP/INDEL in hi-MHS (A)/homoeolog (C) pairs and control-MHS (A)/homoeolog (C) pairs. **(b)** A similar comparison between subgenome A and subgenome D. The number of SNP/indel was normalized by dividing to the overlapped length between MHS and its homoeolog. *** $p < 0.001$, Mann-Whitney U test.

In the section of “Loss of MHS-related sequences after polyploidization” in Results, we conducted similar comparative analysis between subgenome A and subgenomes C/D using *F. iinumae* as the diploid ancestor of both C and D. We identified 7,328 subgenome A MHSs without a subgenome C homoeolog, of which 52% (3,795/7,328) were found to have homoeologous sequences in *F. iinumae*. In comparison, 854 of the 2,791 subgenome C MHSs (29%) without a subgenome A homoeolog were found to have homoeologous sequences in *F. vesca*. Similarly, we identified 7,393 subgenome A MHSs without a subgenome D homoeolog, of which 52% (3,874/7,393) were found to have homoeologous sequences in *F. iinumae*. In Comparison, 891 of the 3,091 subgenome D MHSs (30%) without a subgenome A homoeolog were found to have homoeologous sequences in *F. vesca*. Collectively, these results confirmed that fewer MHSs were lost from subgenome A after polyploidization compared to subgenome B/C/D (29% vs 43%, 29% vs 52%, 30 vs 52%, $p < 2.2e-16$, chi-square test).

In the section of “Frequencies of MHS fragmentation associated with subgenomes A and B” in Results, we identified 1,019/1,041 subgenome C/D MHSs associated with fragmoeologs in subgenome A. We investigated if each fragmentation occurred after polyploidization using *F. iinumae* as the diploid progenitor for both C and D subgenomes. We found that 21% (218/1,019) of subgenome C MHSs fragmented in subgenome A aligned with a continuous sequence in the *F. vesca* genome, indicating that these sequences were likely fragmented after polyploidization. We identified 1,239 subgenome A MHSs associated with fragmoeologs in subgenome C, 44% (549/1,239) of them aligned with a continuous sequence in the *F. iinumae* genome. Similarly, 21% (221/1,041) of subgenome D MHSs aligned with a continuous sequence in the *F. vesca* genome, and 44% (549/1,246) subgenome A MHSs fragmented in subgenome D aligned with a continuous sequence in the *F. iinumae* genome. Collectively, these results show that more MHSs in subgenomes B/C/D have become fragmented after polyploidization compared to subgenome A.

We have added these new analyses in the revised manuscript.

(3) In Discussion part, the relationship between chromatin accessibility and A subgenome dominance should be thoroughly discussed.

Response: We have added discussion on the relationship between chromatin accessibility and subgenome A dominance in the revised manuscript.

(4) Line 480-481 showed ‘The average TPM value of each gene was calculated from the three biological replicates’, but Line 448 showed ‘Two biological replicates of young trifoliolate leaves were collected from seedlings’. Please check the number of biological replicates.

Response: We thank the Reviewer for catching this. We collected tissues of three biological replicates. We conducted RNA-seq data from all three biological replicates, but conducted MH-seq using only two of the three biological replicates. We have clarified this in the revised manuscript.

(5) Line 266, ‘1.5-times lower expression levels’, the ‘1.5-times’ and ‘lower’ seems contradictory.

(6) Line 72, ‘(Fragaria x ananassa)’, the ‘x’ should be ‘×’.

(7) Line 78, ‘Fragaria vesca ssp. bracteata’, the ‘ssp.’ shouldn’t italic.

(8) Line 427-428, ‘Bracheata’ should be ‘bracheata’.

Response: We have modified the text accordingly.

Response to comments from Reviewer #3

This manuscript entitled “Divergence and fragmentation of cis-regulatory sequences and subgenome dominance in octoploid strawberry” by Fang et al. generated a genome-wide map of accessible chromatin regions (ACRs) in cultivated strawberry using MHS sequencing, and compared them within the four subgenomes A, B, C, and D. Their data demonstrated the biased evolution of these MHS sites. I have some concerns and questions.

1. If I understand correctly, the authors identified syntenic MHS sites or genes by only comparing them within the four subgenomes. I think this is not strict. Based on my experience, the authors should identify a closely relative outgroup to examine whether these homologous MHS sites or genes have orthologs or syntelogs in the outgroup. For example, sorghum has been used as the outgroup for comparing the two subgenomes in maize. If the syntenic MHS sites or genes are identified within the same genome, some of the homologous pairs may be generated by segmental duplication or other duplication events. Not all of them are generated by polyploidization.

Response: We appreciate this comment. We investigated the possibility if some of the “syntenic MHSs or genes” between two subgenomes were generated by segmental or other duplication events. We chose *F. iinumae* as the outgroup because it is independent of subgenome A, thus, the syntenic MHSs or gene pairs between *F. iinumae* and subgenome A will not be affected by segmental duplications or other duplication events. For example, if a subgenome A MHS has a syntenic sequence in subgenome B and also has a syntenic region in *F. iinumae*, it would suggest that the syntenic MHS pair between subgenome A and B was only associated with the polyploidization event.

We aligned the sequences of 21,911 subgenome A MHSs, which have syntenic sequences in subgenome B, to the *F. iinumae* reference genome to detect the syntenic regions using MCSscanX. Only 3% (744/21,912) of these MHSs do not have syntenic regions in *F. iinumae*. Similarly, a total of 17,046 syntenic gene pairs were identified between subgenomes A and B. We aligned the CDS sequences of the 17,406 subgenome A genes with the genome of *F. iinumae*. We found that only 5% (920/17,046) of the genes do not have syntenic regions in *F. iinumae*. Additionally, we cannot exclude the possibilities that deletions and other rearrangement events may occur in *F. iinumae* since the independent evolution between *F. iinumae* and polyploid strawberry. Therefore, we conclude that the vast majority of the syntenic MHSs or genes between subgenomes A and B are true syntenic rather than deriving from segmental or other duplication events.

2. The description of the paper is not very clear. For example, how many genes and MHS sites in subgenomes A, B, C, and D? How many of these genes and MHS sites have homoeologous pairs in the genome, and how many of them are singletons? Please describe this kind of information at the beginning of your results.

Response: Following the Reviewer’s comment we have developed a Table S1 that includes all the suggested data.

3. Results, page 5, lines 133-135, “For each homoeologous gene pair, one copy was defined to

have an elevated expression if it expressed at least 1.5-fold higher than the other copy.”. Normally people use 2-fold higher to tell the expression dominance. Why did the authors use a lower standard (1.5-fold higher) here?

Response: We chose a 1.5-fold difference as the threshold in order to investigate the impact of MHS differences on transcriptional difference from more homoeologous gene pairs. For example, among the 6,824 subgenome A dominantly expressed genes, only 1,841 genes showed 2-fold higher expression level than the homoeologous genes from subgenome B. By applying the 1.5-fold threshold, a total of 3,310 subgenome A genes were found to show higher expression than homoeologous genes from subgenome B. We analyzed the chromatin accessibility changes of the 1,841 and 3,310 gene pairs, respectively. We noted marked differences in both groups, albeit slightly lower in the group with 3,310 gene pairs (**Figure 5a**). Similar results were also found from the analyses of the homoeologous gene pairs between subgenome A and subgenomes C/D (**Figure 5, b, c**). Thus, choosing the 1.5-fold difference threshold allowed us to see the impact of MHS differences on transcriptional divergence of a significantly large number of homoeologous gene pairs in strawberry.

Figure 5. MNase sensitivity of transcriptionally divergent homoeologous gene pairs between subgenome A and subgenome B (**a**), or subgenome C (**b**), or subgenome D (**c**). Left panel: MNase sensitivity of homoeologous gene pairs with differential expression levels higher than 2-fold. Right panel: MNase sensitivity of homoeologous gene pairs with different expression levels higher than 1.5-fold.

4. Results, page 6, line 144, for the 6,824 dominantly expressed genes from subgenome A, how many of them have chromatin accessibility data?

Response: Thank you for this interesting question that we have never thought about. At least one MHS was found to be associated with nearly 80% (5,421/6,824) of the subgenome A dominantly

expressed genes. This made us wonder what happened with the remaining 1,403 subgenome A genes that exhibit higher expression than their homoeologs but lack MHSs?

A total of 3,310 subgenome A genes showed higher expression levels than their homoeologs in subgenome B. Nearly 23% (660/3,310) of these genes do not have MHSs in subgenome A. We analyzed the chromatin accessibility of these 660 subgenome A genes and their B homoeologs at the genic and 5 kb flanking regions. We observed similar accessibility levels at the proximal and genic regions (**Figure 6a**). Interestingly, in the more distal regions (>1 kb from TSS/TTS), the chromatin accessibility levels of the subgenome A genes are significantly higher than their homoeologs in subgenome B (**Figure 6a**). A similar pattern is observed in comparisons between subgenome A dominantly expressed genes and their homoeologs in subgenome C/D (**Figure 6, b-c**). These results suggest that distal MHSs are likely responsible for the higher expression levels of these subgenome A genes, which appear to lack MHSs. Distal MHSs are prone to be assigned to wrong genes when employing the strategy of arbitrarily associating MHSs with their closest genes. This challenge is possibly addressed in future studies using high-resolution Hi-C data.

Figure 6. MNase sensitivity of dominantly expressed subgenome A genes that are not associated with MHSs. Dominantly expressed subgenome A genes are compared with homoeologous genes from subgenome B (n = 660) (**a**), subgenome C (n = 714) (**b**), and subgenome D (n = 735) (**c**), respectively.

5. Figure 2d and 2e, the colors for subgenomes A and C are too close.

Response: We have modified the colors to ensure better visualization of individual subgenome.

6. Results, page 7, lines 175-178, this part is confusing. When the authors talked about 2,034 pairs of control-MHSs, did the authors mean the two homoeologous regions show similarity and both are MHSs? I guess my question for the whole paper is when the authors tried to identify MHS pairs, did they only consider the sequence homology or in addition to the sequence similarity, both regions are MHS sites (ACRs)? I am asking this question because we detected many ACRs in one subgenome have homoeologous sequences in the duplicated subgenome, but lost chromatin accessibility, which means one is an ACR, its homoeologous is not.

Response: Thank you again for this interesting question that we have never thought about. These 2,034 pairs of subgenome A control-MHSs and their subgenome B homeologs were selected because the chromatin accessibility level between each pair is not significantly different. However, we did not investigate whether each subgenome B homoeolog is identified as an ACR.

We analyzed the chromatin accessibility of each of the 2,034 pairs of A control-MHSs/B homoeologs. We confirmed that all control-MHSs and their B homoeologs show similar levels of chromatin accessibility (**Figure 7a**). However, only 71% (1,443/2,034) of the B homoeologs are identified as MHSs in subgenome B. The remaining 591 subgenome B homoeologs are below the computational threshold and cannot be called as MHSs, although their MNase sensitivity signals are still clearly visible (**Figure 7a**). By contrast, we observed distinctly different levels of chromatin accessibility of the 2,034 subgenome A hi-MHSs compared to their subgenome B homoeologs (**Figure 7b**). Therefore, the comparative analysis of hi-MHS/homoeolog vs. control-MHS/homoeolog is technically valid.

Figure 7. Comparison of chromatin accessibility between subgenome A MHSs and their subgenome B homoeologs. **(a)** Comparison between subgenome A control-MHSs and their subgenome B homoeologs. A dashed line indicates the computational threshold to be called as MHSs (ACRs). **(b)** Comparison between subgenome A hi-MHSs and their subgenome B homoeologs.

7. Results, page 7, lines 179-181, “The numbers of SNPs/INDELs between hi-MHSs and their homoeologs were significantly greater than those between control-MHSs and their homoeologs (Figure 3d).” This data is kind of surprising. I expected that hi-MHSs are more conserved and should have fewer SNPs/INDELs. How about the SNPs/INDELs between hi-MHSs and lo-MHSs?

Response: The SNPs/INDELs between hi-MHSs and their homoeologs can be classified into two types. The first type of SNPs/INDELs is inherited from their diploid progenitors; the second type of SNPs/INDELs arose after polyploidization. Thus, the first type of SNPs/INDELs indicates divergence of sequences between the two diploid progenitors; whereas the second type of SNPs/INDELs indicates sequence conservation after polyploidization. We demonstrate that the inherited SNPs/INDELs from the progenitor species, rather than mutated SNPs/INDELs after polyploidization, represent the major contribution to the sequence divergence between subgenome A hi-MHSs and their subgenome B homoeologs. Therefore, a similar number of the

second type of SNPs/INDELs would suggest a similar selective constrain between hi-MHSs and control-MHSs after polyploidization.

To further prove this interpretation, we examined the mutated SNPs/INDELs between subgenome A hi-MHSs and their homologs in *F. vesca*. We also examined the mutated SNPs/INDELs between subgenome A control-MHSs and their homologs in *F. vesca*. We found that the average number of SNPs/INDELs between hi-MHSs and their *F. vesca* homologs is similar to the number between control-MHSs and their *F. vesca* homologs (**Figure 8**).

Based on the Reviewer’s comments, we examined the mutated SNPs/INDELs between lo-MHSs and their *F. vesca* homologs. The number of SNPs/INDELs between lo-MHSs and their *F. vesca* homologs is significantly higher than that of hi-MHSs vs their homologs and control-MHSs vs. their homologs (**Figure 8**). Collectively, these results suggest that hi-MHSs and control-MHSs are under a similar selective constraint, while lo-MHSs are under a much looser selective constrain.

Figure 8. Number of SNPs/INDELs between subgenome A MHSs and their homologs in *F. vesca*. *** $p < 0.001$, Mann-Whitney *U* test.

8. Figure 3e, it’s hard to understand. Please add more details in the figure legend.

Response: We have modified the legend, which is now “self-explainable”.

9. Results, page 8, lines 216-217, “These genes were referred to as subgenome A-specific genes.” Do these specific genes have syntelogs in closely related species? My main question is how confident they are all genes.

Response: Based on Reviewer’s concerns, we performed additional analyses to obtain high-confident subgenome A-specific genes. We aligned the genomic sequences of the 7,602 subgenome A-specific genes with the transposable elements (TEs) annotated in the strawberry genome. We retained 7,189 genes that show low sequence similarity with TEs (identity <90 and sequence overlap <80%). We then examined the expressions of these 7,189 genes in multiple

tissues. A total of 4,017 genes were found to be expressed in at least one of the RNA-seq samples (TPM>0), and were classified as high-confidence subgenome A-specific genes. Nearly 33% (1,343/4,017) of these genes have syntelogs in *F. vesca*. We re-performed GO analysis using these 4,017 genes. Genes associated with stress response are again significantly enriched within the high-confident subgenome A-specific genes (**Figure 9**).

Figure 9. GO terms enriched in 4,017 high confident subgenome A-specific genes. BP: biological process; MF: molecular function.

10. Results, page 10, lines 293-294, “We focused on the 3,512 subgenome A MHSs split into two fragmoecologs for further analysis.” How many of these fragmoecologs of the 3,512 subgenome A MHSs are also MHSs?

Response: Among these 3,512 subgenome A MHSs, both fragmoecologs corresponding to 18% (654/3,512) of them are also MHSs; only one fragmoecolog corresponding to 52% (1,815/3,512) of them was an MHS; neither of the two fragmoecologs corresponding to the remaining 30% (1,043/3,512) showed as an MHS.

11. Results, page 11, lines 297-304. The authors wanted to show the point that the fold change of chromatin accessibility between MHSs and their fragmoecologs was positively correlated with the distance between the two fragmoecologs. I guess another important factor is how big the remaining part near the gene side matching the MHSs. I anticipate that the fold change of chromatin accessibility between MHSs and their fragmoecologs is negatively correlated with the length of the retained part of the fragmoecologs. For example, if one fragmoecolog has 90% fragment still matching the homoeologous MHS, the fragmoecolog may still have high chromatin accessibility because some motifs at the fragmoecolog are not disrupted by the insertion.

Response: This is a good and logic comment. We examined the correlation between the

percentage of the retained sequence associated with the proximal fragmoecolog and the fold change of chromatin accessibility between MHSs and their fragmoecologs. The percentage of the retained sequence of the proximal fragmoecolog is calculated by the length of proximal fragmoecolog divided by the length of its corresponding MHS. The MHS/fragmoecologs pairs were categorized in three groups: (1) the sequence percentage of the retained proximal fragmoecolog is high ($\geq 2/3$, G1); (2) the sequence percentage is intermedium (between $1/3$ and $2/3$, G2); (3) the sequence percentage is low ($\leq 1/3$, G3). Although the fold change of chromatin accessibility between MHSs and their fragmoecologs is positively correlated with the sequence percentage of the retained proximal fragmoecologs (**Figure 10**), this correlation was notably weaker compared to the three groups categorized based on the distance between the two fragmoecologs (Figure 6d in the original manuscript).

Figure 10. Impact of the sequence percentage of the retained proximal fragmoecolog on MNase sensitivity change between the MHSs and their fragmented homoeologs. Significant difference is found only between G1 and G3.

12. Results, page 12, section “Transposable elements and MHS fragmentation” and Figure 7a. Are these TEs orientated? If these TEs are orientated into proximal and distal ends depending on the flanking MHSs, do the proximal ends show higher MNase sensitive level?

Response: Thank you for the great comment! The TEs were not orientated. Following this comment, we have oriented each TE based on the positions of the proximal and distal fragmoecologs. We designated the TE terminal adjacent to the distal fragmoecolog as the 5' and the TE terminal adjacent to the proximal fragmoecolog as the 3'. This analysis revealed that the TE terminals adjacent to proximal fragmoecologs show higher MNase sensitivity levels than those adjacent to distal fragmoecologs (**Figure 11**). This new figure is used to replace Figure 7a.

Figure 11. MNase sensitivity levels across TEs (n=232) inserted in MHSs associated with subgenomes B/C/D. The 5' of TE is adjacent to distal fragmoeolog (D); the 3' of TE is adjacent to proximal fragmoeolog (P).

13. Results, page 5, line 134, change “if it expressed” into “if it was expressed”.

14. Results, page 9, line 246, change “sequence” into “sequences”.

Response: We have modified the text accordingly.

References

- Bird, K.A., VanBuren, R., Puzey, J.R., and Edger, P.P. (2018). The causes and consequences of subgenome dominance in hybrids and recent polyploids. *New Phytol* 220, 87-93.
- Deb, S.K., Edger, P.P., Pires, J.C., and McKain, M.R. (2023). Patterns, mechanisms, and consequences of homoeologous exchange in allopolyploid angiosperms: a genomic and epigenomic perspective. *New Phytol* 238, 2284-2304.
- Edger, P.P., Smith, R., McKain, M.R., Cooley, A.M., Vallejo-Marin, M., Yuan, Y.W., Bewick, A.J., Ji, L.X., Platts, A.E., Bowman, M.J., Childs, K.L., Washburn, J.D., Schmitz, R.J., Smith, G.D., Pires, J.C., and Puzey, J.R. (2017). Subgenome dominance in an interspecific hybrid, synthetic allopolyploid, and a 140-year-old naturally established neo-allopolyploid Monkeyflower. *Plant Cell* 29, 2150-2167.
- Edger, P.P., Poorten, T., VanBuren, R., Hardigan, M.A., Colle, M., McKain, M.R., Smith, R.D., Teresi, S., Nelson, A.D.L., Wai, C.M., Alger, E.I., Bird, K.A., Yocca, A.E., Pumplin, N., Ou, S.J., Ben-Zvi, G., Brode, A., Baruch, K., Swale, T., Shiue, L., Acharya, C.B., Cole, G.S., Mower, J.P., Childs, K.L., Jiang, N., Lyons, E., Freeling, M., Puzey, J.R., and Knapp, S.J. (2019). Origin and evolution of the octoploid strawberry genome. *Nat Genet* 51, 541-547.
- Feng, C., Wang, J., Harris, A.J., Folta, K.M., Zhao, M.Z., and Kang, M. (2021). Tracing the diploid ancestry of the cultivated octoploid strawberry. *Mol Biol Evol* 38, 478-485.
- Guo, H., Wang, X.Y., Gundlach, H., Mayer, K.F.X., Peterson, D.G., Scheffler, B.E., Chee, P.W., and Paterson, A.H. (2014). Extensive and biased intergenomic nonreciprocal DNA exchanges shaped a nascent polyploid genome, *Gossypium* (Cotton). *Genetics* 197, 1153-1163.
- Hardigan, M.A., Lorant, A., Pincot, D.D.A., Feldmann, M.J., Famula, R.A., Acharya, C.B., Lee, S., Verma, S., Whitaker, V.M., Bassil, N., Zurn, J., Cole, G.S., Bird, K., Edger, P.P., and

- Knapp, S.J. (2021). Unraveling the complex hybrid ancestry and domestication history of cultivated strawberry. *Mol Biol Evol* 38, 2285-2305.
- Hollister, J.D., and Gaut, B.S. (2009). Epigenetic silencing of transposable elements: A trade-off between reduced transposition and deleterious effects on neighboring gene expression. *Genome Res* 19, 1419-1428.
- Jin, X., Du, H.Y., Zhu, C.M., Wan, H., Liu, F., Ruan, J.W., Mower, J.P., and Zhu, A.D. (2023). Haplotype-resolved genomes of wild octoploid progenitors illuminate genomic diversifications from wild relatives to cultivated strawberry. *Nat Plants* 9, 1252-1266.
- Liston, A., Wei, N., Tennessen, J.A., Li, J.M., Dong, M., and Ashman, T.L. (2020). Revisiting the origin of octoploid strawberry. *Nat Genet* 52, 2-4.
- McRae, L., Beric, A., and Conant, G.C. (2022). Hybridization order is not the driving factor behind biases in duplicate gene losses among the hexaploid Solanaceae. *Proceedings of the Royal Society B* 289, 20221810.
- Sargent, D.J., Yang, Y., Surbanovski, N., Bianco, L., Buti, M., Velasco, R., Giongo, L., and Davis, T.M. (2016). HaploSNP affinities and linkage map positions illuminate subgenome composition in the octoploid, cultivated strawberry (*Fragaria × ananassa*). *Plant Sci* 242, 140-150.
- Session, A.M., and Rokhsar, D.S. (2023). Transposon signatures of allopolyploid genome evolution. *Nature Communications* 14, 3180.
- Tang, H.B., Woodhouse, M.R., Cheng, F., Schnable, J.C., Pedersen, B.S., Conant, G., Wang, X.W., Freeling, M., and Pires, J.C. (2012). Altered patterns of fractionation and exon deletions in *Brassica rapa* support a two-step model of paleohexaploidy. *Genetics* 190, 1563-1574.
- Tennessen, J.A., Govindarajulu, R., Ashman, T.L., and Liston, A. (2014). Evolutionary origins and dynamics of octoploid strawberry subgenomes revealed by dense targeted capture linkage maps. *Genome Biol Evol* 6, 3295-3313.
- Woodhouse, M.R., Cheng, F., Pires, J.C., Lisch, D., Freeling, M., and Wang, X.W. (2014). Origin, inheritance, and gene regulatory consequences of genome dominance in polyploids. *P Natl Acad Sci USA* 111, 5283-5288.
- Zhang, Z.B., Gou, X.W., Xun, H.W., Bian, Y., Ma, X.T., Li, J.Z., Li, N., Gong, L., Feldman, M., Liu, B., and Levy, A.A. (2020). Homoeologous exchanges occur through intragenic recombination generating novel transcripts and proteins in wheat and other polyploids. *P Natl Acad Sci USA* 117, 14561-14571.

Reviewers' Comments:

Reviewer #1:

Remarks to the Author:

While there are some good responses to my first comment, I feel the authors response brings up a number of fundamental problems that cause me to question the biology of this manuscript. Overall I feel the authors have sufficiently responded to my questions about the subgenome identity, however the discussion of the large-scale genome wide homoeologous exchanges is highly disputed (Liston et al 2020, Feng et al 2021, Session and Rokhsar 2023, Jin et al 2023). Please note that while Jin et al find small homoeologous exchanges, they total 20.3 megabases of sequence that do not fit the model proposed in these paragraphs. Discussion of homoeologous exchanges is unnecessary and actually challenges the other results of this manuscript. If there are as many homoeologous exchanges between the subgenomes of allooctoploid strawberry that the authors claim, how is there an ancestral subgenome dominance signal that is maintained in the genome today? The methods listed in the paper (lines 610-626) do not include any information on subgenome assignments being redone based on this massive recombination. The authors are assigning subgenomes based on chromosomes, not on these hypothetical homoeologous exchanges. This language should be removed from the manuscript as it is unsupported, and is counter to the analysis that is actually presented.

The authors seem to misconstrue my second response in order to respond to a strawman so I will be more clear. My argument is not that there are no cases of higher order allopolyploids. The authors cite three ancient allohexaploids that show that the most recent subgenome was not dominant, this is the point of my question not a rebuttal. Not all allotetraploids show subgenome dominance within the first one million years or so (MY), such as *Brachypodium hybridum* (Gordon et al 2020). Since the final strawberry hybridization occurred much more recently than the 5 MY, 10MY, and 52-91MY allohexaploid events cited, it is completely possible that no subgenome dominance mechanism has been established in allooctoploid strawberry. Again, the analysis and data set are significant, but the overall language of subgenome dominance in this organism has not been established in any rigorous manner, including this study.

I do not believe the response to my 3rd comment is sufficient. The order of hybridization was presented based on shared repeat content in Session and Rokhsar 2023 using methods that were first shown to be cytogenetically viable in allotetraploid *Xenopus laevis*, and supports the cytogenetic results in allohexaploid *Camelina sativa* that identify the intermediate allotetraploid ancestor (Mandakova et al 2019). The authors can simply add a line saying they aren't addressing this to make clear, or can include a discussion if they have cytogenetic or other physical evidence that disproves this analysis.

Overall I do not believe the authors have presented these interesting results in an honest and robust manner. They have ignored others' recent advances in strawberry genomics and use language that is inaccurate given the data presented. Observation of cis-regulatory and expression differences in such a young higher order allopolyploid is not evidence of subgenome dominance on its own given that allotetraploids are not guaranteed to establish subgenome dominance in a similar amount of time.

Reviewer #2:

Remarks to the Author:

The authors have revised the manuscript according to reviewers' suggestions.

Reviewer #3:

Remarks to the Author:

The revised manuscript has been improved a lot. I have some further suggestions.

1. The whole paper investigated the subgenome dominance using chromatin accessibility and gene expression data only from one tissue, which is leaf. It is better to examine this phenomenon in multiple tissues. I guess you may get similar results, but data from multiple tissues would make the data more convincing. At this stage, I can accept one tissue.
2. I like the idea that the authors compared A with "B/C/D" together given that there are some inconsistencies on which chromosomes should be assigned to the B/C/D subgenomes. Another way the authors could think of is in addition to comparing with A with "B/C/D" together, they can separate a small dataset that can clearly tell subgenomes A, B, C, and D to see whether the patterns and conclusions are still the same as the conclusions from the comparison of A and "B/C/D".
3. Results, page 5, line 133, "Nearly 68% MHSs were located within ± 1 kb regions of annotated genes (Figure S1b)." I think the proportion 68% contains both "within ± 1 kb regions of annotated genes (only 54%)" and gene bodies.
4. Results, page 6, lines 143-145, add the proportions in the parentheses would be better for people to understand.
5. Results, page 7, lines 175-181, chromosome 7D has the highest average number of MHSs per gene and longer average MHS length per gene (Figure S3). Is it possible that this is due to wrong subgenome classification of these MHSs or genes?
6. Results, page 8, Figure 2b and 2c, you only have *** significance above the A bar, but people do not know which comparison you made to get the significance. A vs B, A vs C, A vs D all showed ***. Please make it clear. Same for Figure 5a.
7. Results, page 11, lines 306-308, "Interestingly, GO analysis of the 7,602 subgenome A-specific genes revealed enrichment of genes associated with plant responses to various biotic and abiotic stresses (Figure 4)." What's the GO enrichment of B/C/D subgenome specific genes? I guess they may be also associated with various biotic and abiotic stresses. This may not be unique for subgenome A-specific genes.
8. Results, page 13, lines 381-384, "A total of 1,307/1,239/1,246 subgenome A MHSs aligned with multiple fragmoeologs from subgenomes B/C/D, respectively. Of these, the majority of subgenome A MHSs (3,512/3,792, 92.6%) were split into two fragmoeologs, while the remaining were split into greater than two fragmoeologs." I may misunderstand. Why this number "3,512" (two fragmoeologs) is bigger than "1,307/1,239/1,246" (multiple fragmoeologs)? Does "A total of 1,307/1,239/1,246 subgenome A MHSs aligned with multiple fragmoeologs from subgenomes B/C/D" not include "split into two fragmoeologs"? If so, why did the authors use "Of these"?
9. Results, page 16, line 470, here should be "the insertion".
10. Results, page 16, lines 477-479, how many cases of reduced chromatin accessibility?

Response to comments from Reviewer 1

While there are some good responses to my first comment, I feel the authors response brings up a number of fundamental problems that cause me to question the biology of this manuscript. Overall I feel the authors have sufficiently responded to my questions about the subgenome identity, however the discussion of the large-scale genome wide homoeologous exchanges is highly disputed (Liston et al 2020, Feng et al 2021, Session and Rokhsar 2023, Jin et al 2023). Please note that while Jin et al find small homoeologous exchanges, they total 20.3 megabases of sequence that do not fit the model proposed in these paragraphs. Discussion of homoeologous exchanges is unnecessary and actually challenges the other results of this manuscript. If there are as many homoeologous exchanges between the subgenomes of allooctoploid strawberry that the authors claim, how is there an ancestral subgenome dominance signal that is maintained in the genome today? The methods listed in the paper (lines 610-626) do not include any information on subgenome assignments being redone based on this massive recombination. The authors are assigning subgenomes based on chromosomes, not on these hypothetical homoeologous exchanges. This language should be removed from the manuscript as it is unsupported, and is counter to the analysis that is actually presented.

Response: We agree, given that there are no analyses of homoeologous exchanges in this manuscript, we have removed that text from the Introduction as suggested.

The authors seem to misconstrue my second response in order to respond to a strawman so I will be more clear. My argument is not that there are no cases of higher order allopolyploids. The authors cite three ancient allohexaploids that show that the most recent subgenome was not dominant, this is the point of my question not a rebuttal. Not all allotetraploids show subgenome dominance within the first one million years or so (MY), such as *Brachypodium hybridum* (Gordon et al 2020). Since the final strawberry hybridization occurred much more recently than the 5 MY, 10MY, and 52-91MY allohexaploid events cited, it is completely possible that no subgenome dominance mechanism has been established in allooctoploid strawberry. Again, the analysis and data set are significant, but the overall language of subgenome dominance in this organism has not been established in any rigorous manner, including this study.

Response: We apologize, clearly, we misunderstood your original comment. We have added additional text to improve the overall language to define subgenome dominance – matching the original definition by Thomas et al. (2006), analyzing gene retention patterns following the At-alpha event in Arabidopsis, and later by Schnabel et al. (2011) on gene retention and expression patterns in maize.

New text added in the Introduction: “The dominant subgenome in maize has lost significantly fewer genes and exhibits higher levels of gene expression (Schnable et al., 2011; Yin et al., 2022) – which are the two primary characteristics used to define subgenome dominance in allopolyploids (Bird et al., 2018).” “Among the four subgenomes (A, B, C, D), the A subgenome, which was contributed by an ancestor most closely related to *Fragaria vesca* ssp. *bracteata*, encodes a significantly greater amount of more highly expressed homoeologs compared to the submissive B/C/D subgenomes (Edger et al., 2019). In addition, the dominant A

subgenome has the lowest TE densities near genes compared to the other three subgenomes (Edger et al., 2019). The dominant A subgenome also lost significantly fewer genes compared to the submissive subgenomes (Edger et al., 2019), similar to patterns reported for Arabidopsis (Thomas et al., 2006) and Maize (Schnable et al., 2011).”

I do not believe the response to my 3rd comment is sufficient. The order of hybridization was presented based on shared repeat content in Session and Rokhsar 2023 using methods that were first shown to be cytogenetically viable in allotetraploid *Xenopus laevis*, and supports the cytogenetic results in allohexaploid *Camelina sativa* that identify the intermediate allotetraploid ancestor (Mandakova et al 2019). The authors can simply add a line saying they aren't addressing this to make clear, or can include a discussion if they have cytogenetic or other physical evidence that disproves this analysis.

Response: The hybridization order as outlined in Session and Rokhsar (2023) is certainly plausible and clearly supported by the analysis of repeats. This is an interesting topic but is outside of the scope of this manuscript and would require additional evidence to further validate or potentially support an alternative model (as you noted above). As you requested, we added a sentence in the final paragraph of the Discussion that hybridization order isn't being investigated as part of this manuscript but should be the focus of a future study. Furthermore, we have added a couple of sentences in the Introduction to outline the proposed hybridization order based on Session and Rokhsar (2023).

New text added in the Introduction: “Lastly, regarding the hybridization order, there are consistent results that the A subgenome donor hybridized with the hexaploid ancestor to form the octoploid as the maternal donor (Njuguna et al., 2013; Edger et al., 2019; Session and Rokhsar, 2023). Based on the analyses of repeat content, Session and Rokhsar (2023) proposed a model by which the tetraploid was formed by the hybridization of the diploid progenitors of C and D, which subsequently hybridized with the B subgenome donor to form the ancestral hexaploid.”

New text added in the Discussion: “Finally, it is worth noting that we did not investigate the impact of hybridization order as part of this study, but it would be worth modeling as part of a future follow-up study.”

Overall I do not believe the authors have presented these interesting results in an honest and robust manner. They have ignored others' recent advances in strawberry genomics and use language that is inaccurate given the data presented. Observation of cis-regulatory and expression differences in such a young higher order allopolyploid is not evidence of subgenome dominance on its own given that allotetraploids are not guaranteed to establish subgenome dominance in a similar amount of time.

Response: We absolutely agree with your statement “... allotetraploids are not guaranteed to establish subgenome dominance in a similar amount of time.” There are many allopolyploids, one of which you noted above, that do not exhibit subgenome-wide expression bias patterns. The first paragraph of the discussion was meant to address this point. However, as you noted here, it

is important that this point is properly addressed in this manuscript. Thus, we have revised the Introduction, including moving text from the discussion to the introduction, to ensure that a reader doesn't misinterpret this important point.

New text added in the Introduction: “Although subgenome dominance has been documented in a number of plant species, not all polyploids exhibit subgenome dominance (Douglas et al., 2015; Sun et al., 2017; Li et al., 2019; Fang et al., 2023). The lack of subgenome dominance was thought to be skewed toward autopolyploids or polyploids with highly similar subgenomes (Garsmeur et al., 2014; Zhao et al., 2017; Alger and Edger, 2020). Thus, the pre-existence of genetic differences among the progenitor species may be driving subgenome dominance in certain allopolyploids. Transposable elements (TEs) content is one genomic feature known to often be highly variable within and among closely related plant species (Qiu et al., 2021; Stitzer et al., 2021; Bozan et al., 2023). Furthermore, it is well known that TEs can impact the expression of neighboring genes genetically and/or epigenetically (Mcclintock, 1956; Feschotte et al., 2002; Hollister and Gaut, 2009; Zhao et al., 2018). The genomic shock from a polyploidization event may induce proliferation of TEs residing in the progenitor genomes, which may further differentiate the TE abundance among different subgenomes. Therefore, it was not surprising that low TE abundance has been previously associated with a dominant subgenome in several allopolyploid species (Schnable et al., 2011; Parkin et al., 2014; Cheng et al., 2016) (Edger et al., 2017; Colle et al., 2019; Edger et al., 2019). In other words, several previous studies have shown that the higher expression level of homoeologs encoded on the dominant subgenome is generally inversely correlated with the density of methylated transposable elements compared to the homoeologs on the submissive subgenomes (Li et al., 2014; Cheng et al., 2016; Edger et al., 2017; Zhao et al., 2017; Liang and Schnable, 2018; Alger and Edger, 2020; Xu et al., 2020; Zhang et al., 2022). To complement these findings, previous studies of polyploids which lack any evidence for subgenome dominance have reported similar TE densities near homoeologs between subgenomes (Douglas et al., 2015; Sun et al., 2017). While differences in TEs have been studied in various allopolyploids, other genomic features, including variation in noncoding regulatory regions, as a potential driver of observed homoeolog expression differences, remain to be investigated in allopolyploids (Alger and Edger, 2020).”

Lastly, we certainly did not intend to give the impression that we were “ignoring others' recent advances in strawberry genomics”. Given that there is general consensus across the community of which chromosomes are assigned to the A and B subgenomes, and here we aim to investigate subgenome expression dominance, we largely focused on comparing the A subgenome to the B subgenomes and A and B/C/D subgenomes. In other words, our analyses are largely centric to comparing the A to the other three subgenomes. In the Introduction, we do acknowledge how recent analyses associated with chromosome assignments for C and D differ for chromosomes 2, 5 and 6 per Hardigan et al. (2023) and Session and Rokhsar (2023).

During the last revision, we included new analyses that involved swapping these chromosomes between the C and D subgenomes. Taking one combination as an example, we re-assigned chromosome 2C as chromosome 2D and chromosome 2D as chromosome 2C (**Figure 1**). We re-calculated and compared the average MHS number across subgenome A, B, and the newly created subgenome C and D. After performing the analysis for all 15 combinations, we found that the average MHS number of subgenome D is still not significantly higher than subgenome B

or C (**Figure 1**), but is significantly lower than subgenome A. Similarly, we also examined the average MHS length per gene and the average number of reads per gene among the four subgenomes in the same 15 combinations. We obtained similar results that the numbers from subgenome D are not significantly higher than those from subgenome B or C, but are significantly lower than subgenome A. In summary, swapping these chromosomes between the C and D subgenomes yielded similar results compared to subgenome A.

Figure 1. Average number of MHSs per gene and average MHS length per gene in each subgenome. Purple boxes indicate the chromosomes from subgenome C and cyan boxes indicate chromosomes from subgenome D. In (a) and (b), we exchanged the chromosome 2, 5, and 6 between subgenome C and D. The total number of MHSs and the total MHS lengths were normalized in the four subgenomes by dividing to their total number of genes from each subgenome. Means that do not share a letter are significantly different ($p < 0.01$, one-way ANOVA with Games-Howell *post-hoc* test).

Response to comments from Reviewer 3

The revised manuscript has been improved a lot. I have some further suggestions.

1. The whole paper investigated the subgenome dominance using chromatin accessibility and gene expression data only from one tissue, which is leaf. It is better to examine this phenomenon in multiple tissues. I guess you may get similar results, but data from multiple tissues would make the data more convincing. At this stage, I can accept one tissue.

Response: We agree with this comment. Unfortunately, we haven't done similar MH-seq experiments using other tissues. We plan to add new datasets in our renewal proposal that will be due at the end of 2024.

2. I like the idea that the authors compared A with "B/C/D" together given that there are some inconsistencies on which chromosomes should be assigned to the B/C/D subgenomes. Another way the authors could think of is in addition to comparing with A with "B/C/D" together, they can separate a small dataset that can clearly tell subgenomes A, B, C, and D to see whether the

patterns and conclusions are still the same as the conclusions from the comparison of A and “B/C/D”.

Response: We thank for this suggestion. We selected a dataset containing only the undisputed chromosomes 1, 4, and 7 of subgenome A, B, C, and D. We compared the average number of MHSs per gene and the average MHS length per gene between subgenome A and combined “B/C/D”. The patterns and conclusions are the same as the conclusions from the comparison of A and “B/C/D” (Figure 2).

Figure 2. Chromatin accessibility of chromosome 1, 4, and 7. The average number of MHSs per gene (a) and the average MHS length per gene (b) were compared between subgenome A and combined subgenomes B, C, and D. These combined values were normalized by dividing to the total number of genes from subgenome B, C, and D. *** $p < 0.001$, Mann-Whitney U test.

3. Results, page 5, line 133, “Nearly 68% MHSs were located within ± 1 kb regions of annotated genes (Figure S1b).” I think the proportion 68% contains both “within ± 1 kb regions of annotated genes (only 54%)” and gene bodies.

Response: Yes, the Reviewer is correct. We have modified the description in the revised manuscript.

4. Results, page 6, lines 143-145, add the proportions in the parentheses would be better for people to understand.

Response: We have added the proportions in the parentheses as the Reviewer suggested.

5. Results, page 7, lines 175-181, chromosome 7D has the highest average number of MHSs per gene and longer average MHS length per gene (Figure S3). Is it possible that this is due to wrong subgenome classification of these MHSs or genes?

Response: We have double-checked this result and did not find any wrong assignment of the MHSs or genes. Chromosome 7A has the greatest numbers of MHSs and genes. However, the average number of MHSs per gene from 7A is not as high as that from 7D (see the table below).

Chromosome 7	A	B	C	D
Number of MHSs	3073	2687	2514	2878
Number of genes	3356	3168	3078	2913
Average MHS per gene	0.92	0.85	0.82	0.99

6. Results, page 8, Figure 2b and 2c, you only have *** significance above the A bar, but people do not know which comparison you made to get the significance. A vs B, A vs C, A vs D all showed ***. Please make it clear. Same for Figure 5a.

Response: We have added the significance mark *** for all comparisons (A vs B, A vs C, A vs D) in both Figure 2 and Figure 5.

7. Results, page 11, lines 306-308, “Interestingly, GO analysis of the 7,602 subgenome A-specific genes revealed enrichment of genes associated with plant responses to various biotic and abiotic stresses (Figure 4).” What’s the GO enrichment of B/C/D subgenome specific genes? I guess they may be also associated with various biotic and abiotic stresses. This may not be unique for subgenome A-specific genes.

Response: We thank the Reviewer for this valuable comment. We identified 6,764, 6,526, and 6,230 genes specific to B, C, and D subgenomes, respectively, and performed similar GO analyses. A total of 74 significantly enriched GO terms related to biological processes (BP) were found from analysis based on the subgenome B-specific genes. Only two GO terms (“jasmonic acid and ethylene-dependent systemic resistance”, and “response to mechanical stimulus”) are related to stress response. Similarly, we identified 8 and 3 significantly enriched BP GO terms from subgenomes C- and D-specific genes, respectively, but none of them are related to stress response. We have added these results in the revised manuscript.

8. Results, page 13, lines 381-384, “A total of 1,307/1,239/1,246 subgenome A MHSs aligned with multiple fragmoeologs from subgenomes B/C/D, respectively. Of these, the majority of subgenome A MHSs (3,512/3,792, 92.6%) were split into two fragmoeologs, while the remaining were split into greater than two fragmoeologs.” I may misunderstand. Why this number “3,512” (two fragmoeologs) is bigger than “1,307/1,239/1,246” (multiple fragmoeologs)? Does “A total of 1,307/1,239/1,246 subgenome A MHSs aligned with multiple fragmoeologs from subgenomes B/C/D” not include “split into two fragmoeologs”? If so, why did the authors use “Of these”?

Response: The text has been modified as follows: A total of 3,792 subgenome A MHSs aligned with multiple fragmoeologs from subgenomes B/C/D (1,307 with B; 1,239 with C; 1,246 with D), respectively. The majority of these subgenome A MHSs (3,512/3,792, 92.6%) were split into two fragmoeologs, while the remaining were split into greater than two fragmoeologs.

9. Results, page 16, line 470, here should be “the insertion”.

Response: We have modified the accordingly.

10. Results, page 16, lines 477-479, how many cases of reduced chromatin accessibility?

Response: Among the 232 cases in which a portion of the inserted TE was integrated into the cognate MHS, 11 MHSs (5%) showed an elevated level of chromatin accessibility compared to the fragmoeologs, 17 MHSs (7%) showed a reduced level of chromatin accessibility, and 204 MHSs (88%) showed a similar level of chromatin accessibility levels compared to their fragmoeologs (see table below). We conducted a comparison of chromatin accessibility levels between 1,970 MHSs and their fragmoeologs broken by TE insertions. Among these 1,970 cases, only 19 (1%) MHSs showed an increased level of chromatin accessibility of the fragmoeologs, 744 (38%) MHSs showed a reduced level of reduced chromatin accessibility, and 1207 (61%) exhibit a similar level of chromatin accessibility compared to their fragmoeologs (see table below).

	Total number	Increased*	Reduced*	Similar
Fragmoeologs caused by TE insertion and the TE was integrated into the cognate MHS	232	11 (5%)	17 (7%)	204 (88%)
Fragmoeologs caused by TE insertion	1,970	19 (1%)	744 (38%)	1,207 (61%)

* The chromatin accessibility level of the MHS is significantly “increased” or “reduced” compared to the cognate fragmoeologs.

References

- Alger, E.I., and Edger, P.P.** (2020). One subgenome to rule them all: underlying mechanisms of subgenome dominance. *Curr Opin Plant Biol* **54**, 108-113.
- Bird, K.A., VanBuren, R., Puzey, J.R., and Edger, P.P.** (2018). The causes and consequences of subgenome dominance in hybrids and recent polyploids. *New Phytol* **220**, 87-93.
- Bozan, I., Achakkagari, S.R., Anglin, N.L., Ellis, D., Tai, H.H., and Strömviik, M.V.** (2023). Pangenome analyses reveal impact of transposable elements and ploidy on the evolution of potato species. *P Natl Acad Sci USA* **120**, e2211117120.
- Cheng, F., Sun, C., Wu, J., Schnable, J., Woodhouse, M.R., Liang, J.L., Cai, C.C., Freeling, M., and Wang, X.W.** (2016). Epigenetic regulation of subgenome dominance following whole genome triplication in *Brassica rapa*. *New Phytol* **211**, 288-299.

- Colle, M., Leisner, C.P., Wai, C.M., Ou, S., Bird, K.A., Wang, J., Wisecaver, J.H., Yocca, A.E., Alger, E.I., Tang, H.B., Xiong, Z.Y., Callow, P., Ben-Zvi, G., Brodt, A., Baruch, K., Swale, T., Shiue, L., Song, G.Q., Childs, K.L., Schillmiller, A., Vorsa, N., Buell, C.R., VanBuren, R., Jiang, N., and Edger, P.P. (2019). Haplotype-phased genome and evolution of phytonutrient pathways of tetraploid blueberry. *Gigascience* **8**, 1-15.
- Douglas, G.M., Gos, G., Steige, K.A., Salcedo, A., Holm, K., Josephs, E.B., Arunkumar, R., Agren, J.A., Hazzouri, K.M., Wang, W., Platts, A.E., Williamson, R.J., Neuffer, B., Lascoux, M., Slotte, T., and Wright, S.I. (2015). Hybrid origins and the earliest stages of diploidization in the highly successful recent polyploid *Capsella bursa-pastoris*. *P Natl Acad Sci USA* **112**, 2806-2811.
- Edger, P.P., Smith, R., McKain, M.R., Cooley, A.M., Vallejo-Marin, M., Yuan, Y.W., Bewick, A.J., Ji, L.X., Platts, A.E., Bowman, M.J., Childs, K.L., Washburn, J.D., Schmitz, R.J., Smith, G.D., Pires, J.C., and Puzey, J.R. (2017). Subgenome dominance in an interspecific hybrid, synthetic allopolyploid, and a 140-year-old naturally established neo-allopolyploid Monkeyflower. *Plant Cell* **29**, 2150-2167.
- Edger, P.P., Poorten, T., VanBuren, R., Hardigan, M.A., Colle, M., McKain, M.R., Smith, R.D., Teresi, S., Nelson, A.D.L., Wai, C.M., Alger, E.I., Bird, K.A., Yocca, A.E., Pumphlin, N., Ou, S.J., Ben-Zvi, G., Brode, A., Baruch, K., Swale, T., Shiue, L., Acharya, C.B., Cole, G.S., Mower, J.P., Childs, K.L., Jiang, N., Lyons, E., Freeling, M., Puzey, J.R., and Knapp, S.J. (2019). Origin and evolution of the octoploid strawberry genome. *Nat Genet* **51**, 541-547.
- Fang, C., Yang, M.Y., Tang, Y.C., Zhang, L., Zhao, H.N., Ni, H.J., Chen, Q.S., Meng, F.L., and Jiang, J.M. (2023). Dynamics of *cis*-regulatory sequences and transcriptional divergence of duplicated genes in soybean. *P Natl Acad Sci USA* **120**, e2303836120.
- Feschotte, C., Jiang, N., and Wessler, S.R. (2002). Plant transposable elements: Where genetics meets genomics. *Nat Rev Genet* **3**, 329-341.
- Garsmeur, O., Schnable, J.C., Almeida, A., Jourda, C., D'Hont, A., and Freeling, M. (2014). Two evolutionarily distinct classes of paleopolyploidy. *Mol Biol Evol* **31**, 448-454.
- Hardigan, M.A., Feldmann, M.J., Pincot, D.A.D., Famula, R.A., Vachev, M.V., Madera, M.A., Zerbe, P., Mars, K., Peluso, P., Rank, D., Ou, S.J., Saski, C.A., Acharya, C.B., Cole, G.S., Yocca, A.E., Platts, A.E., Edger, P.P., and Knapp, S.J. (2023). Blueprint for phasing and assembling the genomes of heterozygous polyploids: application to the octoploid genome of strawberry. *bioRxiv*, doi: 10.1101/2021.1111.1103.467115.
- Hollister, J.D., and Gaut, B.S. (2009). Epigenetic silencing of transposable elements: A trade-off between reduced transposition and deleterious effects on neighboring gene expression. *Genome Res* **19**, 1419-1428.
- Li, A.L., Liu, D.C., Wu, J., Zhao, X.B., Hao, M., Geng, S.F., Yan, J., Jiang, X.X., Zhang, L.Q., Wu, J.Y., Yin, L.J., Zhang, R.Z., Wu, L., Zheng, Y.L., and Mao, L. (2014). mRNA and small RNA transcriptomes reveal insights into dynamic homoeolog regulation of allopolyploid heterosis in nascent hexaploid wheat. *Plant Cell* **26**, 1878-1900.
- Li, Q.H., Qiao, X., Yin, H., Zhou, Y.H., Dong, H.Z., Qi, K.J., Li, L.T., and Zhang, S.L. (2019). Unbiased subgenome evolution following a recent whole-genome duplication in pear (*Pyrus bretschneideri* Rehd.). *Hortic Res* **6**, 34.

- Liang, Z.K., and Schnable, J.C.** (2018). Functional divergence between subgenomes and gene pairs after whole genome duplications. *Mol Plant* **11**, 388-397.
- McClintock, B.** (1956). Controlling elements and the gene. *Cold Spring Harb Sym* **21**, 197-216.
- Njuguna, W., Liston, A., Cronn, R., Ashman, T.L., and Bassil, N.** (2013). Insights into phylogeny, sex function and age of *Fragaria* based on whole chloroplast genome sequencing. *Mol Phylogenet Evol* **66**, 17-29.
- Parkin, I.A.P., Koh, C., Tang, H.B., Robinson, S.J., Kagale, S., Clarke, W.E., Town, C.D., Nixon, J., Krishnakumar, V., Bidwell, S.L., Deneud, F., Belcram, H., Links, M.G., Just, J., Clarke, C., Bender, T., Huebert, T., Mason, A.S., Pires, J.C., Barker, G., Moore, J., Walley, P.G., Manoli, S., Batley, J., Edwards, D., Nelson, M.N., Wang, X.Y., Paterson, A.H., King, G., Bancroft, I., Chalhoub, B., and Sharpe, A.G.** (2014). Transcriptome and methylome profiling reveals relics of genome dominance in the mesopolyploid *Brassica oleracea*. *Genome Biol* **15**, R77.
- Qiu, Y.J., O'Connor, C.H., Della Coletta, R., Renk, J.S., Monnahan, P.J., Noshay, J.M., Liang, Z.K., Gilbert, A., Anderson, S.N., McGaugh, S.E., Springer, N.M., and Hirsch, C.N.** (2021). Whole-genome variation of transposable element insertions in a maize diversity panel. *G3-Genes Genomes Genetics* **11**, jkab238.
- Schnable, J.C., Springer, N.M., and Freeling, M.** (2011). Differentiation of the maize subgenomes by genome dominance and both ancient and ongoing gene loss. *P Natl Acad Sci USA* **108**, 4069-4074.
- Session, A.M., and Rokhsar, D.S.** (2023). Transposon signatures of allopolyploid genome evolution. *Nature Communications* **14**, 3180.
- Stitzer, M.C., Anderson, S.N., Springer, N.M., and Ross-Ibarra, J.** (2021). The genomic ecosystem of transposable elements in maize. *Plos Genet* **17**, e1009768.
- Sun, H.H., Wu, S., Zhang, G.Y., Jiao, C., Guo, S.G., Ren, Y., Zhang, J., Zhang, H.Y., Gong, G.Y., Jia, Z.C., Zhang, F., Tian, J.X., Lucas, W.J., Doyle, J.J., Li, H.Z., Fei, Z.J., and Xu, Y.** (2017). Karyotype stability and unbiased fractionation in the paleo-allotetraploid *Cucurbita* genomes. *Mol Plant* **10**, 1293-1306.
- Thomas, B.C., Pedersen, B., and Freeling, M.** (2006). Following tetraploidy in an Arabidopsis ancestor, genes were removed preferentially from one homeolog leaving clusters enriched in dose-sensitive genes. *Genome Res* **16**, 934-946.
- Xu, W.F., Zhang, Q., Yuan, W., Xu, F.Y., Aslam, M.M., Miao, R., Li, Y., Wang, Q.W., Li, X., Zhang, X., Zhang, K., Xia, T.Y., and Cheng, F.** (2020). The genome evolution and low-phosphorus adaptation in white lupin. *Nature Communications* **11**, 1069.
- Yin, L.W., Xu, G., Yang, J.L., and Zhao, M.X.** (2022). The heterogeneity in the landscape of gene dominance in maize is accompanied by unique chromatin environments. *Mol Biol Evol* **39**, msac198.
- Zhang, Y.Y., Li, Z.J., Liu, J.Y., Zhang, Y., Ye, L.H., Peng, Y., Wang, H.Y., Diao, H., Ma, Y., Wang, M.Y., Xie, Y.L., Tang, T.F., Zhuang, Y.L., Teng, W., Tong, Y.P., Zhang, W.L., Lang, Z.B., Xue, Y.B., and Zhang, Y.J.** (2022). Transposable elements orchestrate subgenome-convergent and -divergent transcription in common wheat. *Nature Communications* **13**.
- Zhao, D.Y., Hamilton, J.P., Vaillancourt, B., Zhang, W.L., Eizenga, G.C., Cui, Y.H., Jiang, J.M., Buell, C.R., and Jiang, N.** (2018). The unique epigenetic features of Pack-MULEs and their impact on chromosomal base composition and expression spectrum. *Nucleic Acids Res* **46**, 2380-2397.

Zhao, M.X., Zhang, B.A., Lisch, D., and Ma, J.X. (2017). Patterns and consequences of subgenome differentiation provide insights into the nature of paleopolyploidy in plants. *Plant Cell* **29**, 2974-2994.

Reviewers' Comments:

Reviewer #1:

Remarks to the Author:

I find the authors responses to comments sufficient and am recommending acceptance without further revision.

Reviewer #3:

Remarks to the Author:

My comments have been well addressed.